# Elliptic anisotropy measurement of the f$_0$(980) hadron in proton-lead collisions and evidence for its quark-antiquark composition

**The CMS Collaboration**\* ✉

Despite the f$_0$(980) hadron having been discovered half a century ago, the question about its quark content has not been settled: it might be an ordinary quark-antiquark (q$\overline{\text{q}}$) meson, a tetraquark (q$\overline{\text{q}}$q$\overline{\text{q}}$) exotic state, a kaon-antikaon (K$\overline{\text{K}}$) molecule, or a quark-antiquark-gluon (q$\overline{\text{q}}$g) hybrid. This paper reports strong evidence that the f$_0$(980) state is an ordinary q$\overline{\text{q}}$ meson, inferred from the scaling of elliptic anisotropies ($v_2$) with the number of constituent quarks ($n_q$), as empirically established using conventional hadrons in relativistic heavy ion collisions. The f$_0$(980) state is reconstructed via its dominant decay channel f$_0$(980) → $\pi^+\pi^-$, in proton-lead collisions recorded by the CMS experiment at the LHC, and its $v_2$ is measured as a function of transverse momentum ($p_T$). It is found that the $n_q = 2$ (q$\overline{\text{q}}$ state) hypothesis is favored over $n_q = 4$ (q$\overline{\text{q}}$q$\overline{\text{q}}$ or K$\overline{\text{K}}$ states) by 7.7, 6.3, or 3.1 standard deviations in the $p_T < 10, 8$, or 6 GeV/$c$ ranges, respectively, and over $n_q = 3$ (q$\overline{\text{q}}$g hybrid state) by 3.5 standard deviations in the $p_T < 8$ GeV/$c$ range. This result represents the first determination of the quark content of the f$_0$(980) state, made possible by using a novel approach, and paves the way for similar studies of other exotic hadron candidates.

One of the most intriguing puzzles in quantum chromodynamics (QCD), the theory describing the strong interaction, is the phenomenon of confinement. Confinement is the peculiar feature of the QCD color charges that they cannot be separated and are fatefully confined in color-neutral bound states known as hadrons. The mechanism for the color confinement is still not well understood. A hadron can be the usual quark-antiquark (q$\overline{\text{q}}$) meson or three-quark (qqq) baryon, but it has been suggested that there also exist less conventional, "exotic" forms, such as tetraquarks or meson molecules (q$\overline{\text{q}}$q$\overline{\text{q}}$), pentaquarks (q$\overline{\text{q}}$qqq), and dibaryons (qqqqqq)[1–3], where q stands for a constituent quark of any flavor. Studies of these exotic states can significantly advance our understanding of how partons can form bound states and in which configurations. This knowledge is fundamental for a deeper understanding of QCD, especially in its nonperturbative regime[4,5].

Exotic hadrons are expected to be short-lived and decay into ordinary hadrons, making it challenging to decipher their original parton structure. The first evidence of a tetraquark or a molecular state, $X$(3872), was reported by the Belle experiment at KEK[6]. Experiments at the CERN LHC, particularly LHCb, have recently observed several new candidates for tetraquarks and pentaquarks, as discussed, e.g., in ref. 7. Those candidates all involve heavy quarks, which implies that their properties can be calculated in nonrelativistic QCD[8,9].

However, similar calculations are hard to perform for hadrons built of only light quarks, as one has to use relativistic QCD in its nonperturbative regime. In particular, the f$_0$(980) hadron, discovered 50 years ago[10–12], has been hypothesized to be an ordinary q$\overline{\text{q}}$ meson, a tetraquark state, a K$\overline{\text{K}}$ molecule, or a q$\overline{\text{q}}$-gluon hybrid state[13–18]. The suggestion that the f$_0$(980) hadron can be a tetraquark extends

\*A list of authors and their affiliations appears at the end of the paper. ✉e-mail: cms-publication-committee-chair@cern.ch

beyond the ground-state constituent quark model, and studies of such states would impact our understanding of QCD and color confinement. Despite a multitude of experimental and theoretical works, the nature of the $f_0(980)$ state has not yet been established, as is evident from ref. 19 and references therein.

This is where high-energy nuclear collisions may come to the rescue. The collisions of lead-lead (PbPb) ions at the LHC aim to recreate the quark-gluon plasma (QGP), widely believed to be the state of matter prevailing in the early universe, when the temperature and energy density were too high to allow for the formation of hadrons. They offer a universal laboratory to study various aspects of QCD, such as the formation of hadrons from the QGP hadronization. A large number of hadron species, presumably including exotic ones, are abundantly produced during and following the phase transition from the QGP to hadronic matter. Indeed, evidence for the production of the $X(3872)$ exotic state in PbPb collisions was reported by the CMS experiment[20]. The QGP phase transition (hadronization), intimately connected to color confinement, is being extensively studied, both experimentally and theoretically. A viable way to describe hadronization is via the coalescence of quarks, now dressed with gluons over the phase transition, into hadrons. The coalescence model was initially proposed to describe the formation of deuterons in targets exposed to proton beams[21] and is now commonly used to model hadronization in relativistic nuclear collisions[22-26].

In heavy ion collisions, the azimuthal distribution of produced particles is anisotropic. The anisotropy is believed to result from the interactions among quarks and gluons created in these collisions, converting the initial approximately elliptical ("almond-like") overlap region of the colliding nuclei with a nonzero impact parameter into the anisotropy of particle momenta[27]. It is noteworthy that the collision geometry anisotropy is generic and also present in head-on heavy ion collisions, as well as in proton-proton (pp) and proton-nucleus collisions, arising from fluctuations in the distribution of constituents inside the colliding objects[28]. While it initially came as a surprise when momentum anisotropy was first observed in pp[29-32] and proton-lead (pPb)[33-40] collisions, it is by now a well-established fact. This momentum anisotropy of quarks is then inherited by the formed hadrons, thus providing information that can be used to experimentally determine the quark content of the hadrons[41], as explained below. Since the

anisotropy has been established at the LHC energies in PbPb, pPb, and even pp collisions with high multiplicity of particles produced, any of these colliding systems can be used for this type of measurements.

Azimuthal distributions of particles are often described by a Fourier series[42],

$$\frac{dN}{d\phi} \propto 1 + \sum_{n=1}^{\infty} 2v_n \cos[n(\phi - \psi_n)], \qquad (1)$$

where $\phi$ is the azimuthal angle of the particle momentum vector and $\psi_n$ is the azimuthal angle of the $n$th harmonic plane, defined in each event such that $\sum_i \sin[n(\phi_i - \psi_n)] = 0$, where the index $i$ runs over all particles in an event. Details on the reconstruction of the harmonic planes using event observables are given in the Methods section. The coefficients $v_n$, called anisotropic flow parameters, generally depend on the particle transverse momentum ($p_T$) and rapidity ($y$). The $v_2$ coefficient, called the elliptic flow, describes the dominant anisotropic component. The second-order harmonic plane angle $\psi_2$ is an approximation of the azimuthal angle of the reaction plane, which is defined by the line connecting the centers of the colliding nuclei and the beam line.

In the coalescence picture, illustrated in Fig. 1, quarks with close spatial positions and momenta are more likely to combine and, therefore, the anisotropic flow coefficients $v_n$ of the formed hadron inherit those of the parent quarks ($v_{n,q}$). If $n_q$ quarks with approximately equal momenta combine to form a hadron, the resulting azimuthal distribution is then given by

$$\frac{dN_h}{d\phi} \propto \left(\frac{dN_q}{d\phi}\right)^{n_q} \propto \left[1 + \sum_{n=1}^{\infty} 2v_{n,q}(p_T^q)\cos(n[\phi - \psi_n])\right]^{n_q}, \qquad (2)$$

where $p_T^q = p_T/n_q$, and $N_h$ ($N_q$) is the multiplicity of hadrons (quarks). For small values of $v_n$, relevant for the measurement reported in this paper, one can simplify Eq. (2) as

$$v_n(p_T) \approx n_q v_{n,q}(p_T/n_q).$$

This expression is commonly referred to as the number of constituent quarks (NCQ) scaling of the anisotropic flow[43]. The anisotropic flow of

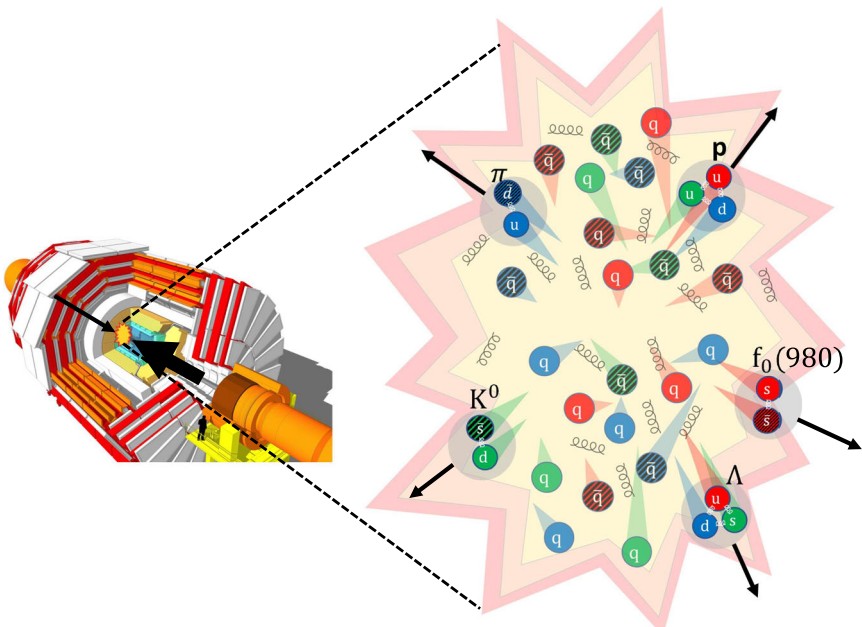

**Fig. 1 | Coalescence hadronization.** This picture illustrates the formation of hadrons in heavy-ion collisions in the coalescence model. Hadrons tend to form when the constituent quarks have similar positions and momenta. [Detector image reprinted from ref. 69, under a CC BY SA 4.0 license].

hadrons formed in heavy ion collisions can therefore reveal the NCQ $n_q$ contained in a hadron, conventional or exotic[41]. Alternatively, such information can also be extracted by measuring the yields and $p_T$ spectra (or their ratios) of these hadrons in heavy ion collisions, albeit in a more model-dependent way[44–47].

The NCQ scaling has been observed to approximately hold for common hadrons in heavy ion collisions at the BNL RHIC[48,49] and at the CERN LHC[38,50,51]. It has also been established in pPb collisions at the LHC by the CMS experiment for $K_S^0$, $\Lambda$, $\Xi^-$, and $\Omega$ hadrons[38,52]. These observations of the NCQ scaling support the validity of the coalescence hadronization model, at least at low $p_T$. We note that the NCQ scaling may additionally arise from other mechanisms in the same or an extended $p_T$ range. The empirical observations of the NCQ scaling do not depend, however, on a particular underlying physics mechanism.

This paper presents the first measurement of the elliptic flow of the $f_0(980)$ state. Data from pPb collisions at a nucleon-nucleon center-of-mass energy of $\sqrt{s_{NN}} = 8.16$ TeV are used. The choice of pPb collisions used in this measurement is driven by the smaller combinatorial background than in PbPb collisions, which simplifies the $f_0(980)$ signal extraction. The elliptic flow coefficient $v_2$ of the $f_0(980)$ state is determined as a function of $p_T$. The NCQ scaling of the $f_0(980)$ hadron $v_2$ coefficient is tested. We demonstrate that the hypothesis of the $f_0(980)$ state being an ordinary $q\bar{q}$ meson is significantly preferred over alternative hypotheses. This novel technique could be used to investigate other exotic hadron candidates. The numeric values from various figures presented in this paper can be found in the HEPData database[53].

## Results
### Analysis of $f_0(980)$ signal
In this paper, the $f_0(980)$ state is measured in pPb collisions at $\sqrt{s_{NN}} = 8.16$ TeV by the CMS experiment. The CMS detector is described in Methods. A high-multiplicity data sample collected in 2016 is used, corresponding to an integrated luminosity of 186 $nb^{-1}$ [54]. The charged-particle multiplicity range is chosen to be $185 \leq N_{trk} < 250$. The $N_{trk}$ multiplicity observable is defined in ref. 52, and its range is chosen to be identical to the one used in that measurement, where significant anisotropic flow has been observed, and to which we compare the NCQ scaling of the $f_0(980)$ measurement. The triggers and event selections are identical to those in ref. 55, as discussed in more detail in the Methods section.

The $f_0(980)$ state is reconstructed within the rapidity $|y| \lesssim 2.4$ via its dominant decay channel, $f_0(980) \to \pi^+\pi^-$ [19]. The pion mass is assigned to all charged-particle tracks. The combinatorial background is modeled via same-charge-sign pion track pairs and subtracted from the opposite-charge-sign dipion mass spectrum. The resulting distribution is then fit to extract the $f_0(980)$ yield. The fit model includes a sum of three Breit–Wigner functions[56–59] corresponding to the $f_0(980)$, $\rho(770)^0$, and $f_2(1270)$ resonances, and a third-order polynomial for the residual background. Details of the fit procedure are described in the Methods section.

The observed elliptic flow $v_2$ of the $f_0(980)$ state is extracted by fitting the yield as a function of $\phi$. The contamination from nonflow correlations—those unrelated to the nuclear collision geometry—is subtracted. After nonflow-contamination subtraction, the elliptic flow coefficient is denoted by $v_2^{sub}$. More details can be found in the Methods section.

In order to compare the $f_0(980)$ $v_2^{sub}$ values to the established NCQ scaling for other hadrons, the $v_{n,q}$ of $K_S^0$, $\Lambda$, $\Xi^-$, and $\Omega$ states measured in the same high-multiplicity range are fit with the following empirical function derived from data:

$$f(KE_T/n_q) = KE_T/n_q \left( p_0 + p_1 KE_T/n_q \right) e^{-p_2 KE_T/n_q}. \quad (3)$$

The argument of the function, $KE_T/n_q$, is related to the kinetic energy per constituent quark, where $KE_T = \sqrt{m^2 + \langle p_t \rangle^2} - m$, $\langle p_t \rangle$ is the average $p_T$ of a $p_T$ bin of the corresponding bound state, and $m$ is its invariant mass. The $f_0(980)$ $\langle p_t \rangle$ values of the $p_T$ bins are obtained from an exponential fit to the $f_0(980)$ candidate $dN/dp_T$ distribution. The $KE_T$ variable is chosen to describe the NCQ scaling as it yields better agreement with the data than $p_T$[49]. The NCQ scaling fit is based on the minimization of the $\chi^2$, assuming that the bin-by-bin uncertainties are uncorrelated, with the coefficients $p_i$ ($i = 0, 1, 2$) being free parameters of the fit. Details about the $n_q$ extraction and about testing of various quark content hypotheses can be found in the Methods section.

### Systematic uncertainties
Systematic uncertainties in the $f_0(980)$ $v_2$ and $v_2^{sub}$ are detailed in the Methods section. The correlation of systematic uncertainties between different $p_T$ bins is taken into account by using a covariance matrix in the $\chi^2$ calculation when extracting $n_q$. The statistical uncertainty in the $f(KE_T/n_q)$ fit is also included as a systematic uncertainty component in the extracted $n_q$. The $n_q$ extraction procedure is repeated for variations in the functional form of $f(KE_T/n_q)$, as well as by using $p_T$ instead of $E_T$ in the NCQ scaling expression given by Eq. (3) (as discussed in the Methods section). The resulting difference in $n_q$ from the default value is taken as the corresponding systematic uncertainty. The systematic uncertainties are listed in Table 1. The uncertainty in the $\langle p_t \rangle$ of the $f_0(980)$ state has a negligible impact on $n_q$.

### Elliptic anisotropy of $f_0(980)$
Figure 2 shows the $v_2^{sub}$ of the $f_0(980)$ state, which is significantly above zero and exhibits a clear trend of rising and then falling with $p_T$,

**Table 1 | Sources and magnitudes of the uncertainties in the extracted $n_q$ of the $f_0(980)$ state in the range $p_T < 10$ GeV/c**

| Source | $n_q$ uncertainty |
|---|---|
| Statistical | 0.16 |
| $f_0(980)$ $v_2$ systematic uncertainty | 0.13 |
| Nonflow effects in $v_2^{sub}$ | 0.04 |
| NCQ scaling fit parameters | 0.02 |
| NCQ scaling fit function | 0.04 |
| NCQ scaling using $p_T/n_q$ | 0.06 |

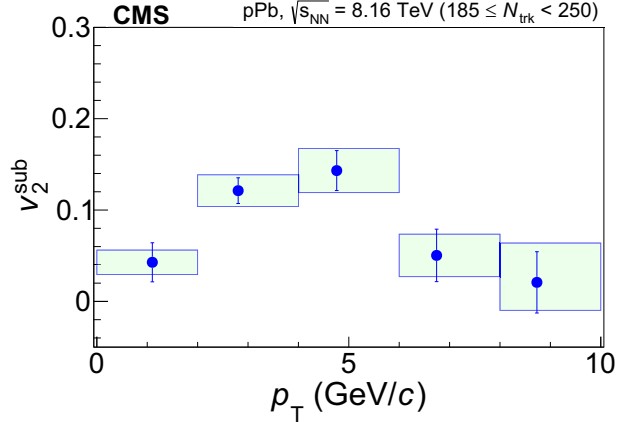

**Fig. 2 | Elliptic anisotropy results.** The nonflow-effect-subtracted elliptic anisotropy $v_2^{sub}$ of the $f_0(980)$ is shown as a function of $p_T$ within $|y| \lesssim 2.4$ in high-multiplicity pPb collisions. The error bars show statistical uncertainties while the shaded areas represent systematic uncertainties.

reaching a maximum in the $4 < p_T < 6$ GeV/$c$ range. Such a trend in $p_T$ has been also observed for other hadrons and is generally considered to come from an interplay between the hydrodynamic expansion at low $p_T$ and partonic energy loss at high $p_T$.

**Quark content of $f_0(980)$**

Figure 3 shows a comparison of $v_2^{sub}/n_q$ for the $f_0(980)$ state with those of $K_S^0$, $\Lambda$, $\Xi^-$, and $\Omega$ hadrons[52] as a function of $KE_T/n_q$. (A similar comparison for $v_2^{sub}/n_q$ as a function of $p_T/n_q$ can be found in the Methods section.) The two sets of the $f_0(980)$ data points correspond to the $n_q = 2$ and 4 hypotheses. The red curve shows the NCQ scaling parameterization of the $v_2^{sub}$ data for these other hadrons (whose $n_q$ values are fixed by their known quark content).

To assess the significance of the result, the log-likelihood ratio $-2\ln(L_{n_q=4}/L_{n_q=2})$ is calculated using the $v_2^{sub}/n_q$ data and the NCQ scaling expectation between the $n_q = 2$ and 4 assumptions. Details about the log-likelihood ratio can be found in the Methods section. The measured value is shown by the red arrow in Fig. 4, together with the distributions of the log-likelihood ratio from pseudo-experiments. The $f_0(980)$ $v_2^{sub}$ values are generated according to the NCQ scaling under the $n_q = 2$ and 4 hypotheses, with a Gaussian smearing to account for

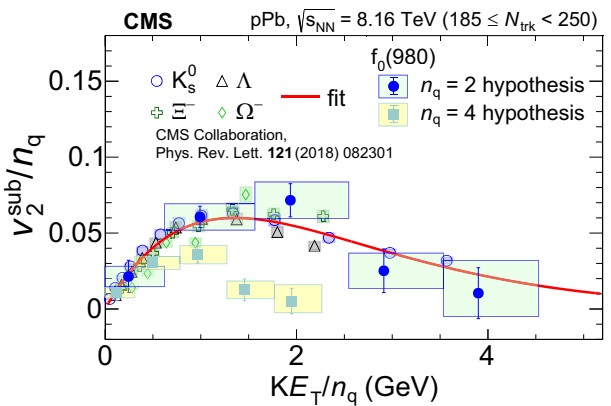

**Fig. 3 | NCQ scaling of elliptic anisotropy.** The $v_2^{sub}/n_q$ of the $f_0(980)$ state (for the $n_q = 2$ and 4 hypotheses) as a function of $KE_T/n_q$, compared with those of $K_S^0$, $\Lambda$, $\Xi^-$, and $\Omega$ strange hadrons[52] in high-multiplicity pPb collisions. The error bars show statistical uncertainties while the shaded areas represent systematic uncertainties. The red curve is the NCQ scaling parameterization of the data for $K_S^0$, $\Lambda$, $\Xi^-$, and $\Omega$ hadrons given by Eq. (3).

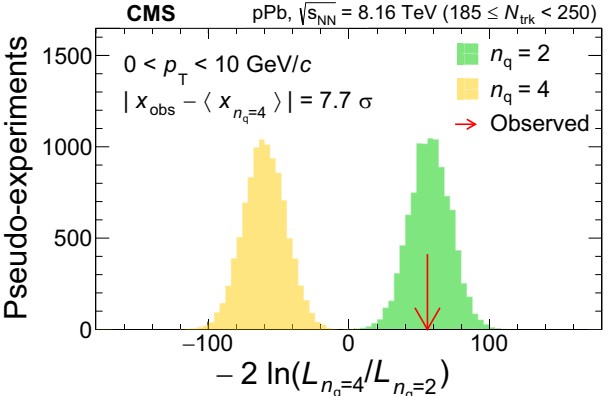

**Fig. 4 | Exclusion significance from $n_q = 4$.** The log-likelihood ratio distributions for the $n_q = 2$ and 4 hypotheses from pseudo-experiments, together with the measured value for the $f_0(980)$ state in the $0 < p_T < 10$ GeV/$c$ range.

the uncertainties. The extracted significance of the $n_q = 2$ hypothesis over the $n_q = 4$ hypothesis is 7.7 standard deviations ($\sigma$) in the $p_T < 10$ GeV/$c$ range. As shown in Fig. 3, the NCQ scaling range as delineated by the $K_S^0$ data extends up to $p_T/n_q$ of 4 GeV/$c$, whereas for the baryons it is restricted to about 2.5 GeV/$c$. For the $n_q = 2$ hypothesis, our high-$p_T$ data start falling out of the measured NCQ scaling $p_T/n_q$ range; for the $n_q = 4$ hypothesis, however, our data are within that range. Consequently, we extract significance values also for two restricted-$p_T$ ranges: $p_T < 8$ and 6 GeV/$c$. The exclusion significances of the $n_q = 4$ vs. 2 hypotheses in these ranges are 6.3 and 3.1$\sigma$, respectively.

The $K\overline{K}$ molecule, if produced by the coalescence of two kaons, would possess the same $v_2$ as that of a tetraquark, and is thus practically also ruled out. It is unclear what $v_2$ a hybrid $q\overline{q}g$ state would attain in pPb collisions because the NCQ scaling has been tested only with ordinary hadrons. If the constituent gluon behaves just like the constituent (anti)quarks, the $v_2$ of a hybrid $q\overline{q}g$ state would scale as $n_q = 3$. Such a state would be ruled out with a 3.5$\sigma$ significance using the $p_T < 8$ GeV/$c$ range, in which the NCQ scaling is adequately measured for the $n_q = 3$ case.

The $\chi^2$ quantity is calculated between the $v_{n,q}$ data of the $f_0(980)$, with floating $n_q$, and the NCQ curve in $KE_T/n_q$ in Fig. 3, with the covariance matrix taking into account correlations among uncertainties. Scans of $\chi^2$ versus $n_q$ are performed, as detailed in the Methods section. Using $f_0(980)$ data within the $p_T < 6$ GeV/$c$ range (a conservative choice, which ensures that the NCQ scaling holds for the $n_q = 2$ hypothesis, given that $p_T/n_q < 3$ GeV/$c$), the preferred $n_q$ value of the $f_0(980)$ is found to be $n_q = 2.40 \pm 0.40$. Assuming the NCQ scaling extends beyond $p_T/n_q \sim 3$ GeV/$c$, the preferred $n_q$ values of $2.10 \pm 0.24$ and $2.07 \pm 0.21$ are extracted in the $p_T < 8$ and 10 GeV/$c$ ranges, respectively. Indeed, the $n_q = 2$ hypothesis for $f_0(980)$ is consistent with the NCQ scaling from the other hadrons, with $\chi^2 = 4.7$ for the 5 data points. Contrary to that, the $n_q = 4$ hypothesis is inconsistent with the data, as evident from the corresponding $\chi^2 = 58$, with a Gaussian $p$ value of $3 \times 10^{-11}$. Consequently, we report a strong evidence for the $q\overline{q}$ quark content of the $f_0(980)$ state.

## Discussion

The $f_0(980)$ state is observed in the $\pi^+\pi^-$ invariant mass distribution of high-multiplicity proton-lead collisions at $\sqrt{s_{NN}} = 8.16$ TeV, using data collected by the CMS experiment in 2016 and corresponding to an integrated luminosity of 186 nb$^{-1}$. The elliptic flow anisotropy $v_2$ of the $f_0(980)$ state is measured as a function of $p_T$ up to 10 GeV/$c$, with respect to the second-order harmonic plane reconstructed from forward/backward energy flow. After subtracting the nonflow contamination, evaluated from $K_S^0$ measurements, we obtain the corrected $v_2^{sub}$ observable. By comparing the $f_0(980)$ $v_2^{sub}$ to those of $K_S^0$, $\Lambda$, $\Xi^-$, and $\Omega$ under the number-of-constituent-quarks scaling hypothesis, we rule out the hypotheses that the $f_0(980)$ is a tetraquark state or a $K\overline{K}$ molecule, in favor of an ordinary $q\overline{q}$ meson hypothesis, at 7.7$\sigma$ (6.3 or 3.1$\sigma$, respectively, if only a restricted range of $p_T < 8$ or 6 GeV/$c$ is considered). The $f_0(980)$ data in the $p_T < 8$ GeV/$c$ range are found to disfavor a quark-antiquark-gluon hybrid state at 3.5$\sigma$. The NCQ of the $f_0(980)$ state, as extracted from a fit to the $v_2^{sub}$ data, is consistent with the value of 2, characteristic of an ordinary meson. Consequently, we find strong evidence that the $f_0(980)$ hadron is a normal quark-antiquark state. We believe that the results reported in this paper offer a solution to a half-century-old puzzle.

The experimental determination of the quark content of the $f_0(980)$ state with high confidence, using this novel approach, is expected to stimulate further experimental investigations as well as theoretical studies. It paves the way for studies of other exotic hadron candidates using the collective flow scaling approach in high-multiplicity proton-nucleus and heavy ion collisions.

## Methods

In this section, we provide experimental details of various steps used in the analysis presented in this paper.

### CMS detector

The central feature of the CMS apparatus is a superconducting solenoid of 6 m internal diameter, providing a magnetic field of 3.8 T. Within the solenoid volume are a silicon pixel and strip tracker, a lead tungstate crystal electromagnetic calorimeter (ECAL), and a brass and scintillator hadron calorimeter (HCAL), each composed of a barrel and two endcap sections. Forward calorimeters extend the pseudorapidity coverage provided by the barrel and endcap detectors. Muons are measured in gas-ionization detectors embedded in the steel flux-return yoke outside the solenoid. The silicon tracker measures charged particles within the range $|\eta| < 2.5$. For nonisolated particles of $1 < p_T < 10$ GeV/$c$ and $|\eta| < 1.4$, the track resolutions are typically 1.5% in $p_T$ and 25–90 (45–150) μm in the transverse (longitudinal) impact parameter[60]. The procedure followed for aligning the detector is described in ref. [61].

### Trigger

Events of interest are selected using a two-tiered trigger system, a suite of triggers based on particle multiplicity. The first level (level-1), composed of custom hardware processors, uses information from the calorimeters and muon detectors to select events at a rate of around 100 kHz within a fixed latency of 4 μs[62]. At level-1, where tracking information is not available, the events were seeded using a tower count in the ECAL and HCAL barrel calorimeters, by selecting events passing a minimum threshold on the number of active towers. An active tower is defined as a trigger tower with a transverse energy exceeding 0.5 GeV. Trigger towers are built by summing energy deposits in the ECAL and HCAL cells in $\Delta\eta \times \Delta\phi = 0.087 \times 0.087$ regions (matching the size of one HCAL cell in the barrel region). The trigger required the tower count to exceed either 115 or 120, depending on the data-taking period.

The second level, known as the high-level trigger, consists of a farm of processors running a version of the full event reconstruction software optimized for fast processing, and reduces the event rate to around 1 kHz before data storage[54]. Several high-level triggers based on the multiplicity of tracks reconstructed either in the pixel detector layers or the full tracker were used for the analysis. The events were first selected requiring more than 125 tracks with $p_T > 0.4$ GeV/$c$, $|\eta| < 2.4$, and the distance of closest approach along the beam axis between the track and the interaction vertex of less than 0.12 cm, reconstructed using only the pixel detector. Further, the events were required to have more than 185 tracks reconstructed in the full tracker with the same $p_T$ and $|\eta|$ requirements, and with the distance of closest approach less than 0.15 cm. The interaction vertex is required to be within 15 cm of the detector center along the beam direction. Offline, we require the number of reconstructed tracks, $N_{trk}$, to be between 185 and 250. The trigger turn-on effect has been shown to have a negligible impact on the result.

More detailed descriptions of the CMS detector, together with a definition of the coordinate system used and the relevant kinematic variables, can be found in refs. [63,64].

### Event selection and reconstruction of the $f_0(980)$ signal

The $f_0(980)$ candidates are reconstructed through the dominant decay channel, $f_0(980) \rightarrow \pi^+\pi^-$ [19]. All charged-particle tracks with $p_T > 0.4$ GeV/$c$ and $|\eta| < 2.4$ passing standard high-purity requirements[60], and with a distance of closest approach to the interaction vertex divided by its uncertainty of less than 3 in both the direction along the beams and in the plane perpendicular to it, are considered as pion candidates. To improve the mass resolution, we only consider tracks with a relative uncertainty in $p_T$ of less than 10%. The $f_0(980)$ candidates are formed from pairs of tracks of opposite-charge-sign, with the charged pion mass assigned to both. The combinatorial background is estimated from same-charge-sign track pairs and is subtracted from the invariant mass spectrum of the $f_0(980)$ candidates. The spectrum is further corrected for the tracking efficiency as a function of the track $p_T$ and $\eta$, as obtained via a HIJING v1.0 simulation[65] followed by the CMS detector response simulation with GEANT4[66]. The analysis is performed in bins of $\phi-\psi_2$, the azimuthal angle of the $f_0(980)$ candidate relative to that of the second harmonic plane. The latter is reconstructed from the energy deposition in the hadron forward (HF) calorimeter covering $3 < \eta < 5$ in the Pb-going direction (resulting in a better resolution compared to that using the opposite HF calorimeter) and corrected for the nonuniform detector performance by using the procedure described in ref. [42]. Figure 5 shows an example of the invariant mass spectrum of the $f_0(980)$ candidates within a $p_T$ range of 4–6 GeV/$c$ and a $\phi-\psi_2$ range of $0-\pi/12$ (where the $\phi-\psi_2$ value is first folded from the full range into the $0-\pi/2$ range to decrease the statistical uncertainty per $\phi - \psi_2$ bin).

Several resonances are evident in the mass spectrum shown in Fig. 5, including a significant $f_0(980)$ peak at ~0.98 GeV/$c^2$. The mass spectrum is fit with a template composed of three Breit–Wigner functions corresponding to the $\rho (770)^0$, $f_0(980)$, and $f_2 (1270)$ resonances, and a third-order polynomial to model the residual background. The fit mass range is chosen to be 0.8–1.7 GeV/$c^2$ in order to exclude the contribution from a $\rho (1700)$ peak at high masses and to avoid the low-mass region (<0.8 GeV/$c^2$) exhibiting a nontrivial turn-on behavior. Since only the right tail of the $\rho (770)^0$ resonance is included in the fit, the extrapolated peak into the lower mass region does not necessarily represent the true shape of the $\rho (770)^0$ resonance. The $\phi$-integrated mass spectrum is fit in each $p_T$ interval to obtain the $f_0(980)$ yield and the line-shapes of the three resonances present within the fit window. The resonant line-shapes are then fixed, and the fit of the mass spectrum is repeated in six individual $\phi-\psi_2$ bins in the corresponding $p_T$ interval, treating the resonance yields as free parameters. The resultant fit to the example $\phi-\psi_2$ bin of the $p_T$ interval is superimposed in Fig. 5, along with the $\chi^2$ of the fit per degree of freedom (dof).

### Extraction of $f_0(980)$ elliptic anisotropy $v_2$ values

Figure 6a shows the $f_0(980)$ yield as a function of $\phi-\psi_2$ in the $4 < p_T < 6$ GeV/$c$ bin as an example. The $f_0(980)$ yield as a function of $\phi-\psi_2$ is fit with Eq. (1) with only the $n = 2$ term to extract the $v_2$ parameter. The fitted $v_2$ values are corrected for the harmonic plane resolution, which represents the precision of the reconstructed $\psi_2$. The resolution is obtained by the three-subevent method[42] and evaluated in each fine multiplicity interval, and an average resolution is obtained weighted by the corresponding $\phi$-integrated yield of $f_0(980)$. The three-subevent method uses the two HFs and the central tracker detector, where the $\eta$ gaps between the subevents help suppress the nonflow effects. Figure 6b shows the corrected $v_2$ of the $f_0(980)$ as a function of $p_T$.

The $v_2$ measurement is contaminated by nonflow correlations, such as back-to-back jet pairs, where an $f_0(980)$ candidate is found within a jet and the harmonic plane is reconstructed from hadrons that include the other fragments of the dijet system. Since $f_0(980)$ is a hadron known to likely contain strange quarks, the relative nonflow contribution $(v_2 - v_2^{sub})/v_2$ to the $f_0(980)$ $v_2$ is assumed to be the same as that for the $K_S^0$ meson, in each $p_T$ bin, where $v_2^{sub}$ represents the elliptic flow after nonflow-effect subtraction. The latter, evaluated using events with low track multiplicity[52], is fit with a second-order polynomial as a function of $p_T$. The relative nonflow contribution to the $f_0(980)$ $v_2$ is evaluated from the fit function at the $\langle p_t \rangle$ in each $p_T$ bin and ranges 9–64% for different $p_T$ bins. The nonflow effects are subtracted to obtain the final $v_2^{sub}$ of the $f_0(980)$ state. The $v_2^{sub}$ of the $f_0(980)$ is shown in Fig. 2 as a function of $p_T$.

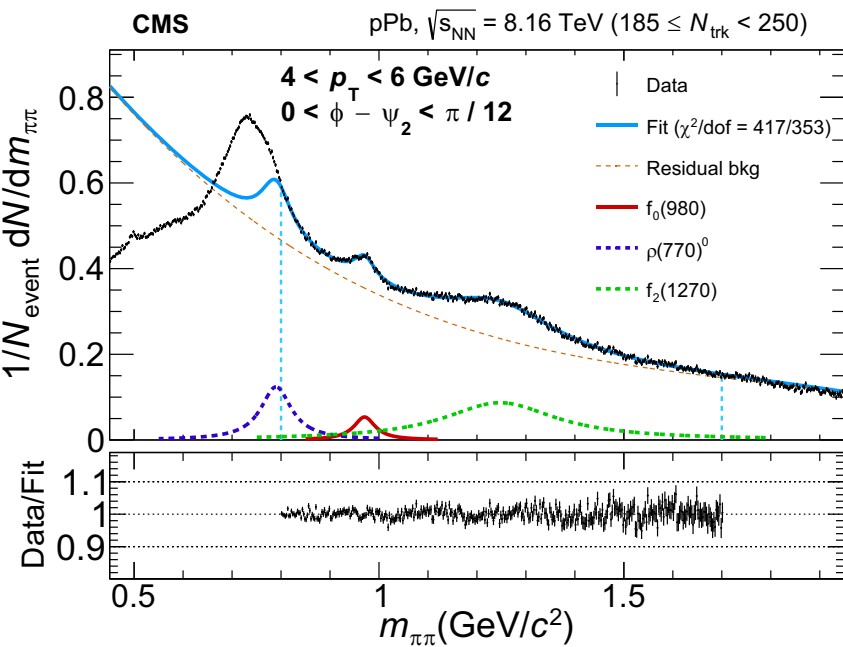

**Fig. 5 | Invariant mass fit.** The invariant mass spectrum of opposite-sign pion pairs after the combinatorial background subtraction, for the pair transverse momentum $4 < p_T < 6$ GeV/$c$ and the azimuthal angle $0 < \phi - \psi_2 < \pi/12$, in high-multiplicity pPb collisions. The solid blue curve is the fit result within the fit range marked with vertical blue dashed lines; the orange dashed curve represents the residual background. The solid red curve represents the $f_0(980)$ signal, while the dashed dark-violet and light-green curves correspond to the background contributions from the $\rho(770)^0$ and $f_2(1270)$ resonances, respectively. The ratio between data and the fit result is shown in the lower panel, with the error bars representing statistical uncertainties only. The low-mass region exhibits a nontrivial turn-on behavior and is not included in the fit.

## Systematic uncertainties in $f_0(980)$ $v_2$ values

The systematic uncertainties in the $f_0(980)$ yield, and consequently those in the $f_0(980)$ $v_2$ values, include those from track selection, track efficiency correction, combinatorial background subtraction, residual background parameterization, resonance line-shape modeling, fit range choice, and the harmonic plane resolution. They are summarized in Table 1 and are evaluated as follows.

- Looser and tighter criteria of track selection are applied, and the obtained $f_0(980)$ $v_2$ results are compared to the default ones, all of which are not corrected for track efficiencies. The uncertainty is 3–22% in the $f_0(980)$ $v_2$ value, depending on $p_T$.
- There could be a difference in the detector response between the same-charge-sign and opposite-charge-sign pairs in the same event. To assess this systematic uncertainty, the default combinatorial background spectrum from same-sign pairs is scaled by the ratio of opposite- to same-sign spectra from mixed events (i.e., when the two tracks forming a pair are taken from different events). The effect on the $f_0(980)$ $v_2$ values is smaller than 3%.
- The residual background is parameterized by second-, fourth-, and fifth-order polynomials besides the default third-order one. The corresponding systematic uncertainty in $v_2$ is found to be less than 5%.
- The resonance mass peaks are alternatively modeled via a relativistic Breit–Wigner function[67] and a relativistic Voigt function[68], which yields a systematic uncertainty in $v_2$ of less than 2%, except in the highest measured $p_T$ interval, where it reaches 25%. The default fit range (0.8–1.7 GeV/$c^2$) is varied by 0.02 GeV/$c^2$ on each side and gives a systematic uncertainty in $v_2$ of less than 8%.
- The statistical uncertainty in the harmonic plane angle extraction is propagated to the $f_0(980)$ $v_2$ and treated as a systematic uncertainty, of ~6%.
- An alternative way to estimate the nonflow contribution to the $f_0(980)$ $v_2$ is by assuming that the absolute nonflow contribution $v_2 - v_2^{\text{sub}}$, instead of the relative $(v_2 - v_2^{\text{sub}})/v_2$ contribution, at a

given $p_T$, is the same as that of the $K_S^0$ meson. The difference between the $f_0(980)$ $v_2^{\text{sub}}$ estimates obtained with the alternative and default methods is treated as the systematic uncertainty in the nonflow-effect subtraction, which is further symmetrized using the larger of the negative and positive variations. The resultant systematic uncertainty band in $v_2^{\text{sub}}$ is then capped between the measured $v_2$ value and zero, ranging from 1% to 33%, depending on $p_T$.
- We have also examined the nonflow contribution using $D^0$ and $\Lambda$ data instead of $K_S^0$ data. Moreover, we have compared the uncorrected $v_2$ distribution of the $f_0(980)$ to those of the $K_S^0, \Lambda, \Xi^-$, and $\Omega$ hadrons. The variations observed in these cross-checks are shown to be fully covered by the systematic uncertainty detailed in the previous item.

These various sources of systematic uncertainties are assumed to be independent of each other. The statistical uncertainty is treated as uncorrelated for different $p_T$ bins. The systematic uncertainty arising from the event plane resolution is assumed to be fully correlated between the $p_T$ bins. For each of the other systematic uncertainties, the $v_2^{\text{sub}}$ covariance matrix element for the $i$th and $j$th $p_T$ bins is calculated as

$$\sigma_{i,j} = \frac{1}{N_{\text{alt}}} \sum_{k=1}^{N_{\text{alt}}} \left( v_{2,k}^{\text{sub}}(p_{T,i}) - v_{2,\text{default}}^{\text{sub}}(p_{T,i}) \right) \left( v_{2,k}^{\text{sub}}(p_{T,j}) - v_{2,\text{default}}^{\text{sub}}(p_{T,j}) \right),$$

$$(4)$$

where $N_{\text{alt}}$ is the number of alternative methods to extract this systematic uncertainty, and $v_{2,\text{default}}^{\text{sub}}$ is the default value. The overall covariance matrix of the $v_2^{\text{sub}}$ is the sum of the covariance matrices corresponding to the various systematic uncertainties. The uncertainty in the $\langle p_t \rangle$ evaluation is estimated using pseudo-experiments and is found to be negligible.

**a**

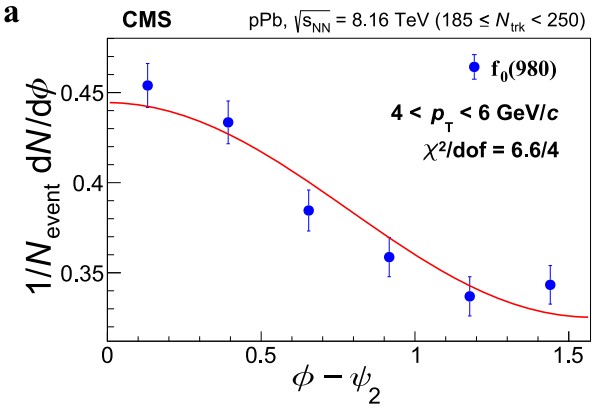

**b**

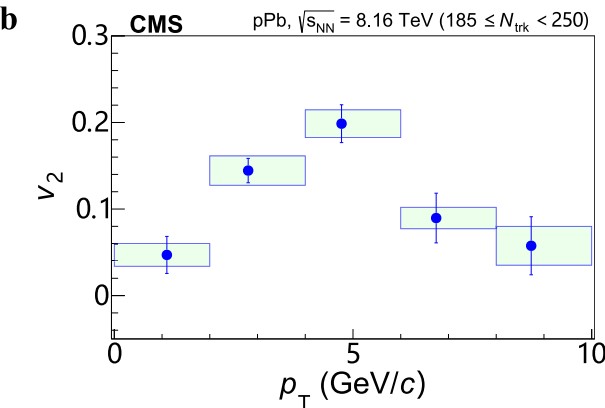

**Fig. 6 | Elliptic anisotropy before the nonflow effect subtraction. a** The $f_0(980)$ yield in the $4 < p_T < 6$ GeV/$c$ range as a function of $\phi-\psi_2$ in high-multiplicity pPb collisions. Error bars show statistical uncertainties. The red curve is a fit to Eq. (1) with only the $n = 2$ term, from which the elliptic anisotropy $v_2$ parameter is extracted. **b** The elliptic anisotropy $v_2$ of the $f_0(980)$ state is shown before the nonflow effect subtraction as a function of $p_T$ within rapidity $|y| \lesssim 2.4$ in high-multiplicity pPb collisions. The error bars show statistical uncertainties while the shaded areas represent systematic uncertainties.

## Cross-checks of the NCQ scaling assumption

The uncertainty used in the parametrization of Eq. (3) of the $v_2^{sub}/n_q$ data for the $K_S^0$, $\Lambda$, $\Xi^-$, and $\Omega^-$ hadrons is the combined statistical and systematic uncertainties of the fit added in quadrature. The resulting best fit parameters are: $p_0 = 0.111 \pm 0.004$, $p_1 = 0.045 \pm 0.017$, and $p_2 = -1.00 \pm 0.08$. The fit yields a relatively large $\chi^2$/dof = 80/34, which indicates that the NCQ scaling is not perfect. To accommodate for this, we increased the uncertainties to achieve the $\chi^2$/dof of 1 and found the effect on the $n_q$ result to be negligible.

The NCQ scaling can also be parameterized in $p_T/n_q$. The fit quality is worse, with $\chi^2$/dof = 170/34. The NCQ-scaled $v_2^{sub}/n_q$ as a function of $p_T/n_q$ is shown in Fig. 7 for the $f_0(980)$ state together with those of the $K_S^0$, $\Lambda$, $\Xi^-$, and $\Omega^-$ hadrons[52]. Using NCQ scaling in $p_T/n_q$, the extracted significance of the $n_q = 2$ hypothesis over the $n_q = 4$ hypothesis is 7.8, 6.2, or $2.4\sigma$, in the $p_T < 10$, 8, or 6 GeV/$c$ ranges, respectively.

To extract the $n_q$ of $f_0(980)$, the $\chi^2$ of the $f_0(980)$ $v_2^{sub}/n_q$ data (denoted by $\vec{y}$) with respect to the NCQ scaling curve (denoted by $\vec{f}$) is calculated by $\chi^2 = (\vec{y} - \vec{f})^T (C_y + C_f)^{-1} (\vec{y} - \vec{f})$, where $C_y$ is the $f_0(980)$ $v_2^{sub}$ covariance matrix scaled by $1/n_q^2$ and $C_f$ is the covariance matrix between the NCQ scaling function from the fit and the $f_0(980)$ $v_2^{sub}/n_q$ data. The latter is given by $C_f = J \cdot C_p \cdot J^T$, where $C_p$ is the covariance matrix of the fit parameters and $J = \partial f / \partial \vec{p}$ is the Jacobian

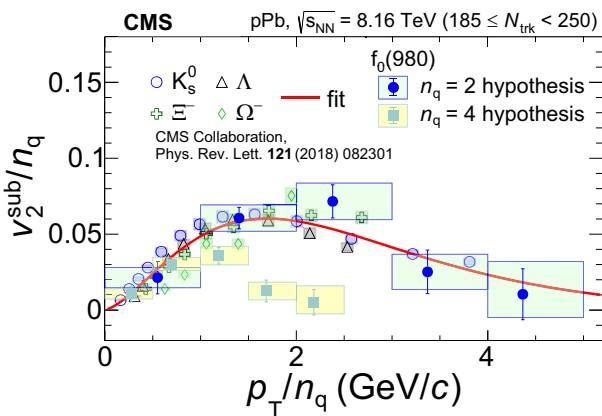

**Fig. 7 | NCQ scaling of elliptic anisotropy in $p_T/n_q$.** The $v_2^{sub}/n_q$ of the $f_0(980)$ state (for the $n_q = 2$ and 4 hypotheses) as a function of $p_T/n_q$ is compared with those of the $K_S^0$, $\Lambda$, $\Xi^-$, and $\Omega^-$ strange hadrons[52] in high-multiplicity pPb collisions. Error bars show the statistical uncertainties while the shaded areas represent systematic uncertainties. The red curve is the NCQ scaling parameterization of the data for the $K_S^0$, $\Lambda$, $\Xi^-$, and $\Omega$ hadrons.

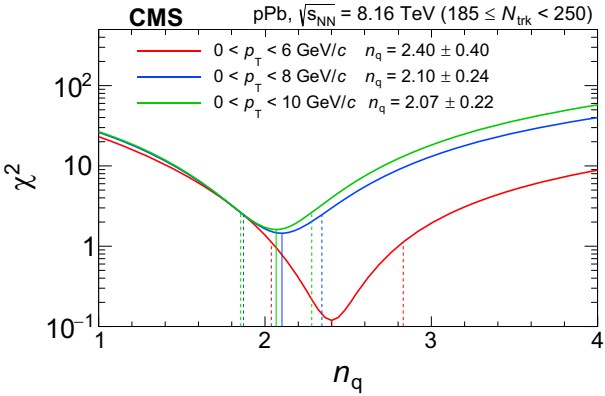

**Fig. 8 | The $\chi^2$ scan.** The $\chi^2$ of the $f_0(980)$ elliptic flow data with respect to the NCQ scaling parameterization, scanned in steps of $n_q$. The three curves correspond to using $f_0(980)$ data for $p_T < 6$, 8, and 10 GeV/$c$, respectively.

matrix that describes how the fit function value changes with the fit parameters $\vec{p}$.

Scans of $\chi^2$ as a function of $n_q$, treated as a continuous parameter, between the $f_0(980)$ $v_2^{sub}/n_q$ data and the NCQ scaling curve, are shown in Fig. 8. The three curves correspond to the $f_0(980)$ data for $p_T < 6$, 8, and 10 GeV/$c$, respectively. The statistical and systematic uncertainties are included in the $\chi^2$ calculation with the covariance matrix. The optimal $n_q$ value is determined at the minimum $\chi^2$, denoted by $\chi^2_{min}$, with the uncertainty bracketed by the $\chi^2_{min} + 1$ level. The corresponding $n_q$ values are listed in Fig. 8, with the uncertainties accounting for the effects of variations in the NCQ fit functional forms and from using $p_T/n_q$ instead of $KE_T/n_q$, which are relatively small (as detailed in Table 1).

Lead-lead collision data from ALICE suggest that the NCQ scaling holds within a precision of 20%[50,51]. We vary the overall fit curve of the NCQ scaling by $\pm 10\%$ and obtain $n_q = 1.95 \pm 0.22$ and $2.19 \pm 0.24$, respectively, for the positive and negative variations, by using $f_0(980)$ data for $p_T < 10$ GeV/$c$. Both these values agree well with the nominal result of $n_q = 2.07 \pm 0.22$, demonstrating the robustness of our result with respect to variations in the NCQ scaling assumption.

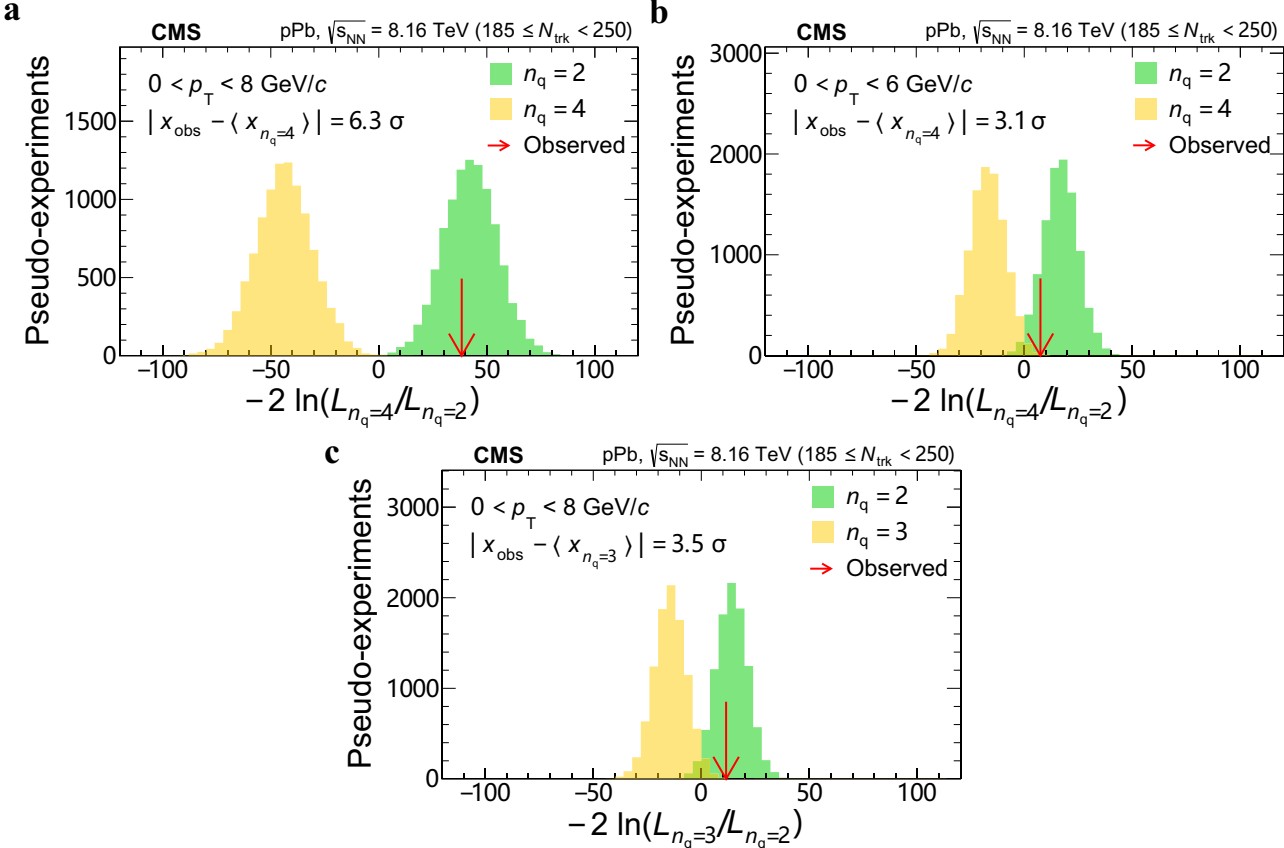

**Fig. 9 | Exclusion significances.** Same as Fig. 4 but using $f_0(980)$ $v_2^{sub}$ data within the restricted ranges $p_T < 8$ GeV/$c$ (**a**) and $p_T < 6$ GeV/$c$ (**b**). **c** The expected log-likelihood ratio distributions for $n_q = 2$ vs. 3 hypotheses from the pseudo-experiments and the observed value for the $f_0(980)$ in data in the $p_T < 8$ GeV/$c$ range to extract the exclusion significance for $n_q = 3$.

The NCQ scaling may be less valid at low $p_T$ where $v_2^{sub}$ likely reflects hydrodynamic behavior, which is mass dependent. However, excluding the lowest $p_T$ data point has negligible effect on our results.

To examine the effects of imperfect NCQ scaling, we carry out further cross-checks as follows. We fit only the $K_S^0$ $v_2^{sub}$ data to obtain an alternative NCQ scaling curve and repeat the analysis. The $f_0(980)$ $n_q$ extracted this way is $2.03 \pm 0.22$. When we use the nominal NCQ scaling curve with $\Lambda$ $v_2^{sub}$ data, the extracted $\Lambda$ $n_q$ value is $2.73 \pm 0.14$. The $2\sigma$ deviation from the nominal value of 3 indicates degree of the validity of the NCQ scaling between the $K_S^0$ and $\Lambda$ hadrons. Similarly, when we use only the $\Lambda$ $v_2^{sub}$ data to obtain the NCQ scaling curve, the extracted $n_q$ is $2.30 \pm 0.22$ for the $f_0(980)$ and $2.29 \pm 0.18$ for the $K_S^0$ states. We have also tested the NCQ scaling validity on $\Omega$ data by using the $K_S^0$, $\Lambda$, and $\Xi^-$ data for the NCQ scaling fit; the extracted $n_q$ value for $\Omega$ is $3.21 \pm 0.69$.

**Exclusion significance determination**

The log-likelihood ratio $-2\ln(L_{n_q=4}/L_{n_q=2})$, evaluated as the $\chi^2$ difference between the $n_q = 2$ and 4 hypotheses, is calculated for the $f_0(980)$ $v_2^{sub}$ data. We also generate pseudo-data of $f_0(980)$ $v_2^{sub}$ according to the NCQ scaling curve for a given $n_q$ hypothesis. The $v_2^{sub}$ uncertainties are taken into account by smearing $v_2^{sub}$ with a Gaussian distribution according to the covariance matrix given by Eq. (4). The corresponding log-likelihood ratio is calculated for the pseudo-data in the same way as for pPb data. The pseudo-experiments yield an expected distribution of the log-likelihood ratio for the two given $f_0(980)$ $n_q$ hypotheses. Each of these distributions is fit with a Gaussian function and the significance of the main hypothesis over the alternative one is extracted as the distance between the Gaussian mean of the alternative distribution (the yellow histogram in, e.g., Fig. 4) and the measured value in data (the red arrow in the same figure), divided by the width of the Gaussian function. The consistency with the main hypothesis can be inferred in a similar way by comparing the value obtained in data with the Gaussian mean and width of the main distribution (the green histogram in the same figure).

The log-likelihood ratio distributions for the $n_q = 2$ vs. 4 hypotheses for the two restricted $p_T$ ranges are shown in Fig. 9a, b. The log-likelihood ratio distributions for the $n_q = 2$ vs. 3 hypotheses in the $p_T < 8$ GeV/$c$ range are shown in Fig. 9c.

## Data availability

Release and preservation of data used by the CMS Collaboration as the basis for publications is guided by the CMS data preservation, re-use and open access policy.

## Code availability

The CMS core software is publicly available in our GitHub repository.

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

## Acknowledgements

We congratulate our colleagues in the CERN accelerator departments for the excellent performance of the LHC and thank the technical and administrative staffs at CERN and at other CMS institutes for their contributions to the success of the CMS effort. In addition, we gratefully acknowledge the computing centers and personnel of the Worldwide LHC Computing Grid and other centers for delivering so effectively the computing infrastructure essential to our analyzes. Finally, we acknowledge the enduring support for the construction and operation of the LHC, the CMS detector, and the supporting computing infrastructure provided by the following funding agencies: SC (Armenia), BMBWF and FWF (Austria); FNRS and FWO (Belgium); CNPq, CAPES, FAPERJ, FAPERGS, and FAPESP (Brazil); MES and BNSF (Bulgaria); CERN; CAS, MoST, and NSFC (China); MINCIENCIAS (Colombia); MSES and CSF (Croatia); RIF (Cyprus); SENESCYT (Ecuador); MoER, ERC PUT and ERDF (Estonia); Academy of Finland, MEC, and HIP (Finland); CEA and CNRS/IN2P3 (France); SRNSF (Georgia); BMBF, DFG, and HGF (Germany); GSRI (Greece); NKFIH (Hungary); DAE and DST (India); IPM (Iran); SFI (Ireland); INFN (Italy); MSIP and NRF (Republic of Korea); MES (Latvia); LAS (Lithuania); MOE and UM (Malaysia); BUAP, CINVESTAV, CONACYT, LNS, SEP, and UASLP-FAI (Mexico); MOS (Montenegro); MBIE (New Zealand); PAEC (Pakistan); MES and NSC (Poland); FCT (Portugal); MESTD (Serbia); MCIN/AEI and PCTI (Spain); MOSTR (Sri Lanka); Swiss Funding Agencies (Switzerland); MST (Taipei); MHESI and NSTDA (Thailand); TUBITAK and TENMAK (Turkey); NASU (Ukraine); STFC (United Kingdom); DOE and NSF (USA). Individuals have received support from the Marie-Curie program and the European Research Council and Horizon 2020 Grant, contract Nos. 675440, 724704, 752730, 758316, 765710, 824093, and COST Action CA16108 (European Union); the Leventis Foundation; the Alfred P. Sloan Foundation; the Alexander von Humboldt Foundation; the Science Committee, project no. 22rl-037 (Armenia); the Belgian Federal Science Policy Office; the Fonds pour la Formation à la Recherche dans l'Industrie et dans l'Agriculture (FRIA-Belgium); the Agentschap voor Innovatie door Wetenschap en Technologie (IWT-Belgium); the F.R.S.-FNRS and FWO (Belgium) under the "Excellence of Science – EOS" – be.h project n. 30820817; the Beijing Municipal Science & Technology Commission, No. Z191100007219010 and Fundamental Research Funds for the Central Universities (China); the Ministry of Education, Youth and Sports (MEYS) of the Czech Republic; the Shota Rustaveli National Science Foundation, grant FR-22-985 (Georgia); the Deutsche Forschungsgemeinschaft (DFG), under Germany's Excellence Strategy – EXC 2121 "Quantum Universe" – 390833306, and under project number 400140256 - GRK2497; the Hellenic Foundation for Research and Innovation (HFRI), Project Number 2288 (Greece); the Hungarian Academy of Sciences, the New National Excellence Program - ÚNKP, the NKFIH research grants K 124845, K 124850, K 128713, K 128786, K 129058, K 131991, K 133046, K 138136, K 143460, K 143477, 2020-2.2.1-ED-2021-00181, and TKP2021-NKTA-64 (Hungary); the Council of Science and Industrial Research, India; ICSC – National Research Center for High Performance Computing, Big Data and Quantum Computing, funded by the EU NexGeneration program (Italy); the Latvian Council of Science; the Ministry of Education and Science, project no. 2022/WK/14, and the National Science Center, contracts Opus 2021/41/B/ST2/01369 and 2021/43/B/ST2/01552 (Poland); the Fundação para a Ciência e a Tecnologia, grant CEECIND/01334/2018 (Portugal); the National Priorities Research Program by Qatar National Research Fund; MCIN/AEI/10.13039/501100011033, ERDF "a way of making Europe", and the Programa Estatal de Fomento de la Investigación Científica y Técnica de Excelencia María de Maeztu, grant MDM-2017-0765 and Programa Severo Ochoa del Principado de Asturias (Spain); the Chulalongkorn Academic into Its 2nd Century Project Advancement Project, and the National Science, Research and Innovation Fund via the Program Management Unit for Human Resources & Institutional Development, Research and Innovation, grant B37G660013 (Thailand); the Kavli Foundation; the Nvidia Corporation; the SuperMicro Corporation; the Welch Foundation, contract C-1845; and the Weston Havens Foundation (USA).

## Author contributions

All authors have contributed to the publication, being variously involved in the design and the construction of the detectors, in writing software, in calibrating sub-systems and operating the detectors, in acquiring data, and in analyzing the processed data.

## Competing interests

The authors declare no competing interests.

## Additional information

## The CMS Collaboration

A. Hayrapetyan[1], A. Tumasyan[1,2], W. Adam[3], J. W. Andrejkovic[3], T. Bergauer[3], S. Chatterjee[3], K. Damanakis[3], M. Dragicevic[3], P. S. Hussain[3], M. Jeitler[3,4], N. Krammer[3], A. Li[3], D. Liko[3], I. Mikulec[3], J. Schieck[3,4], R. Schöfbeck[3], D. Schwarz[3], M. Sonawane[3], S. Templ[3], W. Waltenberger[3], C.-E. Wulz[3,4], M. R. Darwish[5,6], T. Janssen[5], P. Van Mechelen[5], E. S. Bols[7], J. D'Hondt[7], S. Dansana[7], A. De Moor[7], M. Delcourt[7], H. El Faham[7], S. Lowette[7], I. Makarenko[7], D. Müller[7], S. Tavernier[7], M. Tytgat[7,8], G. P. Van Onsem[7], S. Van Putte[7], D. Vannerom[7], B. Clerbaux[9], A. K. Das[9], G. De Lentdecker[9], H. Evard[9], L. Favart[9], P. Gianneios[9], D. Hohov[9], J. Jaramillo[9], A. Khalilzadeh[9], F. A. Khan[9], K. Lee[9], M. Mahdavikhorrami[9], A. Malara[9], S. Paredes[9], L. Thomas[9], M. Vanden Bemden[9], C. Vander Velde[9], P. Vanlaer[9], M. De Coen[8], D. Dobur[8], Y. Hong[8], J. Knolle[8], L. Lambrecht[8], G. Mestdach[8], K. Mota Amarilo[8], C. Rendón[8], A. Samalan[8], K. Skovpen[8], N. Van Den Bossche[8], J. van der Linden[8], L. Wezenbeek[8], A. Benecke[10], A. Bethani[10], G. Bruno[10], C. Caputo[10], C. Delaere[10], I. S. Donertas[10], A. Giammanco[10], Sa. Jain[10], V. Lemaitre[10], J. Lidrych[10], P. Mastrapasqua[10], K. Mondal[10], T. T. Tran[10], S. Wertz[10], G. A. Alves[11], E. Coelho[11], C. Hensel[11], T. Menezes De Oliveira[11], A. Moraes[11], P. Rebello Teles[11], M. Soeiro[11], W. L. Aldá Júnior[12], M. Alves Gallo Pereira[12], M. Barroso Ferreira Filho[12], H. Brandao Malbouisson[12], W. Carvalho[12], J. Chinellato[12,13], E. M. Da Costa[12], G. G. Da Silveira[12,14], D. De Jesus Damiao[12], S. Fonseca De Souza[12], R. Gomes De Souza[12], J. Martins[12,15], C. Mora Herrera[12], L. Mundim[12], H. Nogima[12], J. P. Pinheiro[12], A. Santoro[12], A. Sznajder[12], M. Thiel[12], A. Vilela Pereira[12], C. A. Bernardes[14,16], L. Calligaris[16], T. R. Fernandez Perez Tomei[16], E. M. Gregores[16], P. G. Mercadante[16], S. F. Novaes[16], B. Orzari[16], Sandra S. Padula[16], A. Aleksandrov[17], G. Antchev[17], R. Hadjiiska[17], P. Iaydjiev[17], M. Misheva[17], M. Shopova[17], G. Sultanov[17], A. Dimitrov[18], L. Litov[18], B. Pavlov[18], P. Petkov[18], A. Petrov[18], E. Shumka[18], S. Keshri[19], S. Thakur[19], T. Cheng[20], T. Javaid[20], L. Yuan[20], Z. Hu[21], J. Liu[21], K. Yi[21,22,271], G. M. Chen[23,24], H. S. Chen[23,24], M. Chen[23,24], F. Iemmi[23], C. H. Jiang[23], A. Kapoor[23,25], H. Liao[23], Z.-A. Liu[23,24], R. Sharma[23,26], J. N. Song[23,24], J. Tao[23], C. Wang[23,24], J. Wang[23], Z. Wang[23,24], H. Zhang[23], A. Agapitos[27], Y. Ban[27], A. Levin[27], C. Li[27], Q. Li[27], Y. Mao[27], S. J. Qian[27], X. Sun[27], D. Wang[27], H. Yang[27], L. Zhang[27], C. Zhou[27], Z. You[28], K. Jaffel[29], N. Lu[29], G. Bauer[22,272], X. Gao[9,30], D. Leggat[30], H. Okawa[30], Z. Lin[31], C. Lu[31], M. Xiao[31], C. Avila[32], D. A. Barbosa Trujillo[32], A. Cabrera[32], C. Florez[32], J. Fraga[32], J. A. Reyes Vega[32], J. Mejia Guisao[33], F. Ramirez[33], M. Rodriguez[33], J. D. Ruiz Alvarez[33], D. Giljanovic[34], N. Godinovic[34], D. Lelas[34], A. Sculac[34], M. Kovac[35], T. Sculac[35], P. Bargassa[36], V. Brigljevic[36], B. K. Chitroda[36], D. Ferencek[36], K. Jakovcic[36], S. Mishra[36], A. Starodumov[36,280], T. Susa[36], A. Attikis[37], K. Christoforou[37], S. Konstantinou[37], J. Mousa[37], C. Nicolaou[37], F. Ptochos[37], P. A. Razis[37], H. Rykaczewski[37], H. Saka[37], A. Stepennov[37], M. Finger[38], M. Finger Jr[38], A. Kveton[38], E. Ayala[39], E. Carrera Jarrin[40], A. A. Abdelalim[41,42,273], E. Salama[41,43,274], M. A. Mahmoud[44], Y. Mohammed[44], K. Ehataht[45], M. Kadastik[45], T. Lange[45], S. Nandan[45], C. Nielsen[45], J. Pata[45], M. Raidal[45], L. Tani[45], C. Veelken[45], H. Kirschenmann[46], K. Osterberg[46], M. Voutilainen[46], S. Bharthuar[47], E. Brücken[47], F. Garcia[47], K. T. S. Kallonen[47], R. Kinnunen[47], T. Lampén[47], K. Lassila-Perini[47], S. Lehti[47], T. Lindén[47], L. Martikainen[47], M. Myllymäki[47], M. M. Rantanen[47], H. Siikonen[47], E. Tuominen[47], J. Tuominiemi[47], P. Luukka[48], H. Petrow[48], M. Besancon[49], F. Couderc[49], M. Dejardin[49], D. Denegri[49], J. L. Faure[49], F. Ferri[49], S. Ganjour[49], P. Gras[49], G. Hamel de Monchenault[49], V. Lohezic[49], J. Malcles[49], J. Rander[49], A. Rosowsky[49],

M. Ö. Sahin [49], A. Savoy-Navarro [49,50], P. Simkina [49], M. Titov [49], M. Tornago [49], C. Baldenegro Barrera [51], F. Beaudette [51], A. Buchot Perraguin [51], P. Busson [51], A. Cappati [51], C. Charlot [51], M. Chiusi [51], F. Damas [51], O. Davignon [51], A. De Wit [51], B. A. Fontana Santos Alves [51], S. Ghosh [51], A. Gilbert [51], R. Granier de Cassagnac [51], A. Hakimi [51], B. Harikrishnan [51], L. Kalipoliti [51], G. Liu [51], J. Motta [51], M. Nguyen [51], C. Ochando [51], L. Portales [51], R. Salerno [51], J. B. Sauvan [51], Y. Sirois [51], A. Tarabini [51], E. Vernazza [51], A. Zabi [51], A. Zghiche [51], J.-L. Agram [52,53], J. Andrea [52], D. Apparu [52], D. Bloch [52], J.-M. Brom [52], E. C. Chabert [52], C. Collard [52], S. Falke [52], U. Goerlach [52], C. Grimault[52], R. Haeberle [52], A.-C. Le Bihan [52], M. Meena [52], G. Saha [52], M. A. Sessini [52], P. Van Hove [52], S. Beauceron [54], B. Blancon [54], G. Boudoul [54], N. Chanon [54], J. Choi [54], D. Contardo [54], P. Depasse [54], C. Dozen [21,54], H. El Mamouni[54], J. Fay [54], S. Gascon [54], M. Gouzevitch [54], C. Greenberg[54], G. Grenier [54], B. Ille [54], I. B. Laktineh[54], M. Lethuillier [54], L. Mirabito[54], S. Perries[54], A. Purohit [54], M. Vander Donckt [54], P. Verdier [54], J. Xiao [54], G. Adamov[55], I. Lomidze [55], Z. Tsamalaidze [55,280], V. Botta [56], L. Feld [56], K. Klein [56], M. Lipinski [56], D. Meuser [56], A. Pauls [56], N. Röwert [56], M. Teroerde [56], S. Diekmann [57], A. Dodonova [57], N. Eich [57], D. Eliseev [57], F. Engelke [57], J. Erdmann [57], M. Erdmann[57], P. Fackeldey [57], B. Fischer [57], T. Hebbeker [57], K. Hoepfner [57], F. Ivone [57], A. Jung [57], M. Y. Lee [57], F. Mausolf [57], M. Merschmeyer [57], A. Meyer [57], S. Mukherjee [57], D. Noll [57], F. Nowotny[57], A. Pozdnyakov [57], Y. Rath[57], W. Redjeb [57], F. Rehm[57], H. Reithler [57], U. Sarkar [57], V. Sarkisovi [57], A. Schmidt [57], A. Sharma [57], J. L. Spah [57], A. Stein [57], F. Torres Da Silva De Araujo [57,58], L. Vigilante[57], S. Wiedenbeck [57], S. Zaleski[57], C. Dziwok [59], G. Flügge [59], W. Haj Ahmad [59,60], T. Kress [59], A. Nowack [59], O. Pooth [59], A. Stahl [59], T. Ziemons [59], A. Zotz [59], H. Aarup Petersen [61], M. Aldaya Martin [61], J. Alimena [61], S. Amoroso[61], Y. An [61], S. Baxter [61], M. Bayatmakou [61], H. Becerril Gonzalez [61], O. Behnke [61], A. Belvedere [61], S. Bhattacharya [61], F. Blekman [61,62], K. Borras [57,61], A. Campbell [61], A. Cardini [61], C. Cheng[61], F. Colombina [61], S. Consuegra Rodríguez [61], G. Correia Silva [61], M. De Silva [61], G. Eckerlin[61], D. Eckstein [61], L. I. Estevez Banos [61], O. Filatov [61], E. Gallo [61,62], A. Geiser [61], A. Giraldi [61], V. Guglielmi [61], M. Guthoff [61], A. Hinzmann [61], A. Jafari [61,63], L. Jeppe [61], N. Z. Jomhari [61], B. Kaech [61], M. Kasemann [61], C. Kleinwort [61], R. Kogler [61], M. Komm [61], D. Krücker [61], W. Lange[61], D. Leyva Pernia [61], K. Lipka [61,64], W. Lohmann [61,65], R. Mankel [61], I.-A. Melzer-Pellmann [61], M. Mendizabal Morentin [61], A. B. Meyer [61], G. Milella [61], A. Mussgiller [61], L. P. Nair [61], A. Nürnberg [61], Y. Otarid[61], J. Park [61], D. Pérez Adán [61], E. Ranken [61], A. Raspereza [61], B. Ribeiro Lopes [61], J. Rübenach[61], A. Saggio [61], M. Scham [57,61,66], S. Schnake [57,61], P. Schütze [61], C. Schwanenberger [61,62], D. Selivanova [61], K. Sharko [61], M. Shchedrolosiev [61], R. E. Sosa Ricardo [61], D. Stafford[61], F. Vazzoler [61], A. Ventura Barroso [61], R. Walsh [61], Q. Wang [61], Y. Wen [61], K. Wichmann[61], L. Wiens [57,61], C. Wissing [61], Y. Yang [61], A. Zimermmane Castro Santos [61], A. Albrecht [62], S. Albrecht [62], M. Antonello [62], S. Bein [62], L. Benato [62], S. Bollweg[62], M. Bonanomi [62], P. Connor [62], M. Eich[62], K. El Morabit [62], Y. Fischer [62], C. Garbers [62], E. Garutti [62], A. Grohsjean [62], J. Haller [62], H. R. Jabusch [62], G. Kasieczka [62], P. Keicher[62], R. Klanner [62], W. Korcari [62], T. Kramer [62], V. Kutzner [62], F. Labe [62], J. Lange [62], A. Lobanov [62], C. Matthies [62], A. Mehta [62], L. Moureaux [62], M. Mrowietz[62], A. Nigamova [62], Y. Nissan[62], A. Paasch [62], K. J. Pena Rodriguez [62], T. Quadfasel [62], B. Raciti [62], M. Rieger [62], D. Savoiu [62], J. Schindler [62], P. Schleper [62], M. Schröder [62], J. Schwandt [62], M. Sommerhalder [62], H. Stadie [62], G. Steinbrück [62], A. Tews[62], M. Wolf [62], S. Brommer [67], M. Burkart[67], E. Butz [67], T. Chwalek [67], A. Dierlamm [67], A. Droll[67], N. Faltermann [67], M. Giffels [67], A. Gottmann [67], F. Hartmann [67,68], R. Hofsaess [67], M. Horzela [67], U. Husemann [67], J. Kieseler [67], M. Klute [67], R. Koppenhöfer [67], J. M. Lawhorn [67], M. Link[67], A. Lintuluoto [67], S. Maier [67], S. Mitra [67], M. Mormile [67], Th. Müller [67], M. Neukum[67], M. Oh [67], E. Pfeffer [67], M. Presilla [67], G. Quast [67], K. Rabbertz [67], B. Regnery [67], N. Shadskiy [67], I. Shvetsov [67], H. J. Simonis [67], M. Toms [67], N. Trevisani [67], R. F. Von Cube [67], M. Wassmer [67], S. Wieland [67], F. Wittig[67], R. Wolf [67], X. Zuo [67], G. Anagnostou[69], G. Daskalakis [69], A. Kyriakis[69], A. Papadopoulos[68,69], A. Stakia [69], P. Kontaxakis [70], G. Melachroinos[70], A. Panagiotou[70], I. Papavergou [70], I. Paraskevas [70], N. Saoulidou [70], K. Theofilatos [70], E. Tziaferi [70], K. Vellidis [70], I. Zisopoulos [70], G. Bakas [71], T. Chatzistavrou[71], G. Karapostoli [71], K. Kousouris [71], I. Papakrivopoulos [71], E. Siamarkou[71], G. Tsipolitis[71], A. Zacharopoulou[71], K. Adamidis[72], I. Bestintzanos[72], I. Evangelou [72], C. Foudas[72], C. Kamtsikis[72], P. Katsoulis[72], P. Kokkas [72], P. G. Kosmoglou Kioseoglou [72], N. Manthos [72], I. Papadopoulos [72], J. Strologas [72], M. Bartók [73,74], C. Hajdu [73], D. Horvath [73,75,275], K. Márton[73], F. Sikler [73], V. Veszpremi [73], M. Csanád [76], K. Farkas [76], M. M. A. Gadallah [76,77], Á. Kadlecsik [76], P. Major [76], K. Mandal [76], G. Pásztor [76], A. J. Rádl [73,76], G. I. Veres [76], P. Raics[78], B. Ujvari [78], G. Zilizi [78], G. Bencze[75], S. Czellar[75], J. Molnar[75], Z. Szillasi[75], T. Csorgo [73,79], F. Nemes [73,79], T. Novak [79], J. Babbar [80], S. Bansal [80], S. B. Beri[80], V. Bhatnagar [80], G. Chaudhary [80], S. Chauhan [80], N. Dhingra [80,81], A. Kaur [80], A. Kaur [80], H. Kaur [80], M. Kaur [80], S. Kumar [80], K. Sandeep [80], T. Sheokand[80], J. B. Singh [80], A. Singla [80], A. Ahmed [82], A. Bhardwaj [82], A. Chhetri [82], B. C. Choudhary [82], A. Kumar [82], A. Kumar [82], M. Naimuddin [82], K. Ranjan [82], S. Saumya [82], S. Baradia [83], S. Barman [83,84], S. Bhattacharya [83], S. Dutta [83],

S. Dutta [83], S. Sarkar[83], M. M. Ameen [85], P. K. Behera [85], S. C. Behera [85], S. Chatterjee [85], P. Jana [85], P. Kalbhor [85], J. R. Komaragiri [85,86], D. Kumar [85,86], P. R. Pujahari [85], N. R. Saha [85], A. Sharma [85], A. K. Sikdar [85], S. Verma [85], S. Dugad[87], M. Kumar [87], G. B. Mohanty [87], P. Suryadevara[87], A. Bala [88], S. Banerjee [88], R. M. Chatterjee[88], R. K. Dewanjee [88,89], M. Guchait [88], Sh. Jain [88], A. Jaiswal[88], S. Karmakar [88], S. Kumar [88], G. Majumder [88], K. Mazumdar [88], S. Parolia [88], A. Thachayath [88], S. Bahinipati [90,91], C. Kar [90], D. Maity [90,92], P. Mal [90], T. Mishra [90], V. K. Muraleedharan Nair Bindhu [90,92], K. Naskar [90,92], A. Nayak [90,92], P. Sadangi[90], P. Saha [90], S. K. Swain [90], S. Varghese [90,92], D. Vats [90,92], S. Acharya [93,94], A. Alpana [93], S. Dube [93], B. Gomber [93,94], B. Kansal [93], A. Laha [93], B. Sahu [93,94], S. Sharma [93], K. Y. Vaish[93], H. Bakhshiansohi [61,63], E. Khazaie [63,95], M. Zeinali [63,96], S. Chenarani [97,98], S. M. Etesami [97], M. Khakzad [97], M. Mohammadi Najafabadi [97], M. Grunewald [99], M. Abbrescia [100,101], R. Aly [42,100,102], A. Colaleo [100,101], D. Creanza [100,102], B. D'Anzi [100,101], N. De Filippis [100,102], M. De Palma [100,101], A. Di Florio [100,102], W. Elmetenawee [42,100,101], L. Fiore [100], G. Iaselli [100,102], M. Louka[100,101], G. Maggi [100,102], M. Maggi [100], I. Margjeka [100,101], V. Mastrapasqua [100,101], S. My [100,101], S. Nuzzo [100,101], A. Pellecchia [100,101], A. Pompili [100,101], G. Pugliese [100,102], R. Radogna [100], G. Ramirez-Sanchez [100,102], D. Ramos [100], A. Ranieri [100], L. Silvestris [100], F. M. Simone [100,101], Ü. Sözbilir [100], A. Stamerra [100], R. Venditti [100], P. Verwilligen [100], A. Zaza [100,101], C. Battilana [103,104], D. Bonacorsi [103,104], L. Borgonovi [103], R. Campanini [103,104], P. Capiluppi [103,104], A. Castro [103,104], F. R. Cavallo [103], M. Cuffiani [103,104], G. M. Dallavalle [103], T. Diotalevi [103,104], F. Fabbri [103], A. Fanfani [103,104], D. Fasanella [103,104], P. Giacomelli [103], L. Giommi [103,104], C. Grandi [103], L. Guiducci [103,104], S. Lo Meo [103,105], L. Lunerti [103,104], S. Marcellini [103], G. Masetti [103], F. L. Navarria [103,104], A. Perrotta [103], F. Primavera [103,104], A. M. Rossi [103,104], T. Rovelli [103,104], G. P. Siroli [103,104], S. Costa [106,107,108], A. Di Mattia [106], R. Potenza [106,107], A. Tricomi [106,107,108], C. Tuve [106,107], P. Assiouras [109], G. Barbagli [109], G. Bardelli [109,110], B. Camaiani [109,110], A. Cassese [109], R. Ceccarelli [109], V. Ciulli [109,110], C. Civinini [109], R. D'Alessandro [109,110], E. Focardi [109,110], T. Kello[109], G. Latino [109,110], P. Lenzi [109,110], M. Lizzo [109], M. Meschini [109], S. Paoletti [109], A. Papanastassiou[109,110], G. Sguazzoni [109], L. Viliani [109], L. Benussi [111], S. Bianco [111], S. Meola [111,112], D. Piccolo [111], P. Chatagnon [113], F. Ferro [113], E. Robutti [113], S. Tosi [113,114], A. Benaglia [115], G. Boldrini [115,116], F. Brivio [115], F. Cetorelli [115], F. De Guio [115,116], M. E. Dinardo [115,116], P. Dini [115], S. Gennai [115], R. Gerosa [115,116], A. Ghezzi [115,116], P. Govoni [115,116], L. Guzzi [115], M. T. Lucchini [115,116], M. Malberti [115], S. Malvezzi [115], A. Massironi [115], D. Menasce [115], L. Moroni [115], M. Paganoni [115,116], D. Pedrini [115], B. S. Pinolini[115], S. Ragazzi [115,116], T. Tabarelli de Fatis [115,116], D. Zuolo [115], S. Buontempo [117], A. Cagnotta [117,118], F. Carnevali[117,118], N. Cavallo [117,119], F. Fabozzi [117,119], A. O. M. Iorio [117,118], L. Lista [117,118,120], P. Paolucci [68,117], B. Rossi [117], C. Sciacca [117,118], R. Ardino [121], P. Azzi [121], N. Bacchetta [121,122], A. Bergnoli [121], M. Biasotto [121,123], D. Bisello [121,124], P. Bortignon [121], G. Bortolato [121,124], A. Bragagnolo [121,124], R. Carlin [121,124], P. Checchia [121], T. Dorigo [121], F. Gasparini [121,124], U. Gasparini [121,124], E. Lusiani [121], M. Margoni [121,124], F. Marini [121], A. T. Meneguzzo [121,124], M. Migliorini [121,124], J. Pazzini [121,124], P. Ronchese [121,124], R. Rossin [121,124], G. Strong [121], M. Tosi [121,124], A. Triossi [121,124], S. Ventura [121], H. Yarar[121,124], M. Zanetti [121,124], P. Zotto [121,124], A. Zucchetta [121,124], S. Abu Zeid [125,274], C. Aimè [125,126], A. Braghieri [125], S. Calzaferri [125], D. Fiorina [125], P. Montagna [125,126], V. Re [125], C. Riccardi [125,126], P. Salvini [125], I. Vai [125,126], P. Vitulo [125,126], S. Ajmal [127,128], G. M. Bilei [127], D. Ciangottini [127,128], L. Fanò [127,128], M. Magherini [127,128], G. Mantovani[127,128], V. Mariani [127,128], M. Menichelli [127], F. Moscatelli [127,129], A. Rossi [127,128], A. Santocchia [127,128], D. Spiga [127], T. Tedeschi [127,128], P. Asenov [130,131], P. Azzurri [130], G. Bagliesi [130], R. Bhattacharya [130], L. Bianchini [130,131], T. Boccali [130], E. Bossini [130], D. Bruschini [130,132], R. Castaldi [130], M. A. Ciocci [130,131], M. Cipriani [130,131], V. D'Amante [130,133], R. Dell'Orso [130], S. Donato [130], A. Giassi [130], F. Ligabue [130,132], D. Matos Figueiredo [130], A. Messineo [130,131], M. Musich [130,131], F. Palla [130], A. Rizzi [130,131], G. Rolandi [130,132], S. R. Chowdhury [130], T. Sarkar [130], A. Scribano [130], P. Spagnolo [130], R. Tenchini [130], G. Tonelli [130,131], N. Turini [130,133], A. Venturi [130], P. G. Verdini [130], P. Barria [134], C. Basile [134,135], M. Campana [134,135], F. Cavallari [134], L. Cunqueiro Mendez [134,135], D. Del Re [134,135], E. Di Marco [134], M. Diemoz [134], F. Errico [134,135], E. Longo [134,135], P. Meridiani [134], J. Mijuskovic [134,135], G. Organtini [134,135], F. Pandolfi [134], R. Paramatti [134,135], C. Quaranta [134,135], S. Rahatlou [134,135], C. Rovelli [134], F. Santanastasio [134,135], L. Soffi [134], N. Amapane [136,137], R. Arcidiacono [136,138], S. Argiro [136,137], M. Arneodo [136,138], N. Bartosik [136], R. Bellan [136,137], A. Bellora [136,137], C. Biino [136], C. Borca [136,137], N. Cartiglia [136], M. Costa [136,137], R. Covarelli [136,137], N. Demaria [136], L. Finco [136], M. Grippo [136,137], B. Kiani [136,137], F. Legger [136], F. Luongo [136,137], C. Mariotti [136], L. Markovic [136,137], S. Maselli [136], A. Mecca [136,137], E. Migliore [136,137], M. Monteno [136], R. Mulargia [136], M. M. Obertino [136,137], G. Ortona [136], L. Pacher [136,137], N. Pastrone [136], M. Pelliccioni [136], M. Ruspa [136,138], F. Siviero [136,137], V. Sola [136,137], A. Solano [136,137], A. Staiano [136], C. Tarricone [136,137], D. Trocino [136], G. Umoret [136,137], E. Vlasov [136,137], S. Belforte [139], V. Candelise [139,140], M. Casarsa [139], F. Cossutti [139], K. De Leo [139,140], G. Della Ricca [139,140], S. Dogra [141], J. Hong [141], C. Huh [141],

B. Kim [141], D. H. Kim [141], J. Kim[141], H. Lee [141], S. W. Lee[141], C. S. Moon [141], Y. D. Oh [141], M. S. Ryu [141], S. Sekmen [141], Y. C. Yang [141], M. S. Kim [142], G. Bak [143], P. Gwak [143], H. Kim [143], D. H. Moon [143], E. Asilar [144], D. Kim [144], T. J. Kim [144], J. A. Merlin[144], S. Choi [145], S. Han[145], B. Hong [145], K. Lee[145], K. S. Lee [145], S. Lee [145], J. Park[145], S. K. Park[145], J. Yoo [145], J. Goh [146], S. Yang [146], H. S. Kim [147], Y. Kim[147], S. Lee[147], J. Almond[148], J. H. Bhyun[148], J. Choi [148], W. Jun [148], J. Kim [148], S. Ko [148], H. Kwon [148], H. Lee [148], J. Lee [148], J. Lee [148], B. H. Oh [148], S. B. Oh [148], H. Seo [148], U. K. Yang[148], I. Yoon [148], W. Jang [149], D. Y. Kang [149], Y. Kang[149], S. Kim [149], B. Ko[149], J. S. H. Lee [149], Y. Lee [149], I. C. Park [149], Y. Roh[149], I. J. Watson [149], S. Ha [150], H. D. Yoo [150], M. Choi [151], M. R. Kim [151], H. Lee [151], Y. Lee[151], I. Yu [151], T. Beyrouthy[152], K. Dreimanis [153], A. Gaile [153], G. Pikurs[153], A. Potrebko [153], M. Seidel [153], V. Veckalns [153], N. R. Strautnieks [154], M. Ambrozas [155], A. Juodagalvis [155], A. Rinkevicius [155], G. Tamulaitis [155], N. Bin Norjoharuddeen [156], I. Yusuff [156,157], Z. Zolkapli [156], J. F. Benitez [158], A. Castaneda Hernandez [158], H. A. Encinas Acosta[158], L. G. Gallegos Maríñez[158], M. León Coello [158], J. A. Murillo Quijada [158], A. Sehrawat [158], L. Valencia Palomo [158], G. Ayala [159], H. Castilla-Valdez [159], H. Crotte Ledesma[159], E. De La Cruz-Burelo [159], I. Heredia-De La Cruz [159,160], R. Lopez-Fernandez [159], C. A. Mondragon Herrera[159], A. Sánchez Hernández [159], C. Oropeza Barrera [161], M. Ramírez García [161], I. Bautista [162], I. Pedraza [162], H. A. Salazar Ibarguen [162], C. Uribe Estrada [162], I. Bubanja[163], N. Raicevic [163], P. H. Butler [164], A. Ahmad [165], M. I. Asghar[165], A. Awais [165], M. I. M. Awan[165], H. R. Hoorani [165], W. A. Khan [165], V. Avati[166], L. Grzanka [166], M. Malawski [166], H. Bialkowska [167], M. Bluj [167], B. Boimska [167], M. Górski [167], M. Kazana [167], M. Szleper [167], P. Zalewski [167], K. Bunkowski [168], K. Doroba [168], A. Kalinowski [168], M. Konecki [168], J. Krolikowski [168], A. Muhammad [168], K. Pozniak [169], W. Zabolotny [169], M. Araujo [170], D. Bastos [170], C. Beirão Da Cruz E Silva [170], A. Boletti [170], M. Bozzo [170], T. Camporesi [170], G. Da Molin [170], P. Faccioli [170], M. Gallinaro [170], J. Hollar [170], N. Leonardo [170], T. Niknejad [170], A. Petrilli [170], M. Pisano [170], J. Seixas [170], J. Varela [170], J. W. Wulff[170], P. Adzic [171], P. Milenovic [171], M. Dordevic [172], J. Milosevic [172], V. Rekovic[172], M. Aguilar-Benitez[173], J. Alcaraz Maestre [173], Cristina F. Bedoya [173], M. Cepeda [173], M. Cerrada [173], N. Colino [173], B. De La Cruz [173], A. Delgado Peris [173], A. Escalante Del Valle [173], D. Fernández Del Val [173], J. P. Fernández Ramos [173], J. Flix [173], M. C. Fouz [173], O. Gonzalez Lopez [173], S. Goy Lopez [173], J. M. Hernandez [173], M. I. Josa [173], D. Moran [173], C. M. Morcillo Perez [173], Á. Navarro Tobar [173], C. Perez Dengra [173], A. Pérez-Calero Yzquierdo [173], J. Puerta Pelayo [173], I. Redondo [173], D. D. Redondo Ferrero [173], L. Romero[173], S. Sánchez Navas [173], L. Urda Gómez [173], J. Vazquez Escobar [173], C. Willmott[173], J. F. de Trocóniz [174], B. Alvarez Gonzalez [175], J. Cuevas [175], J. Fernandez Menendez [175], S. Folgueras [175], I. Gonzalez Caballero [175], J. R. González Fernández [175], E. Palencia Cortezon [175], C. Ramón Álvarez [175], V. Rodríguez Bouza [175], A. Soto Rodríguez [175], A. Trapote [175], C. Vico Villalba [175], P. Vischia [175], S. Bhowmik [176], S. Blanco Fernández [176], J. A. Brochero Cifuentes [176], I. J. Cabrillo [176], A. Calderon [176], J. Duarte Campderros [176], M. Fernandez [176], G. Gomez [176], C. Lasaosa García [176], C. Martinez Rivero [176], P. Martinez Ruiz del Arbol [176], F. Matorras [176], P. Matorras Cuevas [176], E. Navarrete Ramos [176], J. Piedra Gomez [176], L. Scodellaro [176], I. Vila [176], J. M. Vizan Garcia [176], M. K. Jayananda [177], B. Kailasapathy [177,178], D. U. J. Sonnadara [177], D. D. C. Wickramarathna [177], W. G. D. Dharmaratna [179,180], K. Liyanage [179], N. Perera [179], N. Wickramage [179], D. Abbaneo [68], C. Amendola [68], E. Auffray [68], G. Auzinger [68], J. Baechler[68], D. Barney [68], A. Bermúdez Martínez [68], M. Bianco [68], B. Bilin [68], A. A. Bin Anuar [68], A. Bocci [68], C. Botta [68], E. Brondolin [68], C. Caillol [68], G. Cerminara [68], N. Chernyavskaya [68], D. d'Enterria [68], A. Dabrowski [68], A. David [68], A. De Roeck [68], M. M. Defranchis [68], M. Deile [68], M. Dobson [68], L. Forthomme [68], G. Franzoni [68], W. Funk [68], S. Giani[68], D. Gigi[68], K. Gill [68], F. Glege [68], L. Gouskos [68], M. Haranko [68], J. Hegeman [68], B. Huber[68], V. Innocente [68], T. James [68], P. Janot [68], O. Kaluzinska [68], S. Laurila [68], P. Lecoq [68], E. Leutgeb [68], C. Lourenço [68], B. Maier [68], L. Malgeri [68], M. Mannelli [68], A. C. Marini [68], M. Matthewman[68], F. Meijers [68], S. Mersi [68], E. Meschi [68], V. Milosevic [68], F. Monti [68], F. Moortgat [68], M. Mulders [68], I. Neutelings [68], S. Orfanelli[68], F. Pantaleo [68], G. Petrucciani [68], A. Pfeiffer [68], M. Pierini [68], D. Piparo [68], H. Qu [68], D. Rabady [68], G. Reales Gutiérrez[68], M. Rovere [68], H. Sakulin [68], S. Scarfi [68], C. Schwick[68], M. Selvaggi [68], A. Sharma [68], K. Shchelina [68], P. Silva [68], P. Sphicas [68,70], A. G. Stahl Leiton [68], A. Steen [68], S. Summers [68], D. Treille [68], P. Tropea [68], A. Tsirou[68], D. Walter [68], J. Wanczyk [68,181], J. Wang[68], S. Wuchterl [68], P. Zehetner [68], P. Zejdl [68], W. D. Zeuner[68], T. Bevilacqua [182,183], L. Caminada [182,183], A. Ebrahimi [182], W. Erdmann [182], R. Horisberger [182], Q. Ingram [182], H. C. Kaestli [182], D. Kotlinski [182], C. Lange [182], M. Missiroli [182,183], L. Noehte [182,183], T. Rohe [182], T. K. Aarrestad [184], K. Androsov [181,184], M. Backhaus [184], A. Calandri [184], C. Cazzaniga [184], K. Datta [184], A. De Cosa [184], G. Dissertori [184], M. Dittmar[184], M. Donegà [184], F. Eble [184], M. Galli [184], K. Gedia [184], F. Glessgen [184], C. Grab [184], D. Hits [184], W. Lustermann [184], A.-M. Lyon [184], R. A. Manzoni [184], M. Marchegiani [184], L. Marchese [184], C. Martin Perez [184], A. Mascellani [181,184], F. Nessi-Tedaldi [184], F. Pauss [184], V. Perovic [184], S. Pigazzini [184], C. Reissel [184], T. Reitenspiess [184], B. Ristic [184], F. Riti [184], R. Seidita [184], J. Steggemann [181,184], D. Valsecchi [184], R. Wallny [184], C. Amsler [183,185], P. Bärtschi [183],

D. Brzhechko[183], M. F. Canelli[183], K. Cormier[183], J. K. Heikkilä[183], M. Huwiler[183], W. Jin[183], A. Jofrehei[183], B. Kilminster[183], S. Leontsinis[183], S. P. Liechti[183], A. Macchiolo[183], P. Meiring[183], U. Molinatti[183], A. Reimers[183], P. Robmann[183], S. Sanchez Cruz[183], M. Senger[183], F. Stäger[183], Y. Takahashi[183], R. Tramontano[183], C. Adloff[186,187], D. Bhowmik[186], C. M. Kuo[186], W. Lin[186], P. K. Rout[186], P. C. Tiwari[86,186], S. S. Yu[186], L. Ceard[188], Y. Chao[188], K. F. Chen[188], P. S. Chen[188], Z. G. Chen[188], A. De Iorio[188], W.-S. Hou[188], T. H. Hsu[188], Y. W. Kao[188], R. Khurana[188], G. Kole[188], Y. Y. Li[188], R.-S. Lu[188], E. Paganis[188], X. F. Su[188], J. Thomas-Wilsker[188], L. S. Tsai[188], H. Y. Wu[188], E. Yazgan[188], C. Asawatangtrakuldee[189], N. Srimanobhas[189], V. Wachirapusitanand[189], D. Agyel[190], F. Boran[190], Z. S. Demiroglu[190], F. Dolek[190], I. Dumanoglu[190,191], E. Eskut[190], Y. Guler[190,192], E. Gurpinar Guler[190,192], C. Isik[190], O. Kara[190], A. Kayis Topaksu[190], U. Kiminsu[190], G. Onengut[190], K. Ozdemir[190,193], A. Polatoz[190], B. Tali[190,194], U. G. Tok[190], S. Turkcapar[190], E. Uslan[190], I. S. Zorbakir[190], M. Yalvac[195,196], B. Akgun[197], I. O. Atakisi[197], E. Gülmez[197], M. Kaya[197,198], O. Kaya[197,199], S. Tekten[197,200], A. Cakir[201], K. Cankocak[191,201,276], Y. Komurcu[201], S. Sen[201,202], O. Aydilek[60,203], S. Cerci[194,203], V. Epshteyn[203], B. Hacisahinoglu[203], I. Hos[203,204], B. Kaynak[203], S. Ozkorucuklu[203], O. Potok[203], H. Sert[203], C. Simsek[203], C. Zorbilmez[203], B. Isildak[205], D. Sunar Cerci[194,205], A. Boyaryntsev[206], B. Grynyov[206], L. Levchuk[207], D. Anthony[208], J. J. Brooke[208], A. Bundock[208], F. Bury[208], E. Clement[208], D. Cussans[208], H. Flacher[208], M. Glowacki[208], J. Goldstein[208], H. F. Heath[208], L. Kreczko[208], S. Paramesvaran[208], L. Robertshaw[208], S. Seif El Nasr-Storey[208], V. J. Smith[208], N. Stylianou[7,208], K. Walkingshaw Pass[208], R. White[208], A. H. Ball[209], K. W. Bell[209], A. Belyaev[209,210], C. Brew[209], R. M. Brown[209], D. J. A. Cockerill[209], C. Cooke[209], K. V. Ellis[209], K. Harder[209], S. Harper[209], M.-L. Holmberg[208,209], J. Linacre[209], K. Manolopoulos[209], D. M. Newbold[209], E. Olaiya[209], D. Petyt[209], T. Reis[209], A. R. Sahasransu[209], G. Salvi[209], T. Schuh[209], C. H. Shepherd-Themistocleous[209], I. R. Tomalin[209], T. Williams[209], R. Bainbridge[211], P. Bloch[211], C. E. Brown[211], O. Buchmuller[211], V. Cacchio[211], C. A. Carrillo Montoya[211], G. S. Chahal[211,212], D. Colling[211], J. S. Dancu[211], I. Das[211], P. Dauncey[211], G. Davies[211], J. Davies[211], M. Della Negra[211], S. Fayer[211], G. Fedi[211], G. Hall[211], M. H. Hassanshahi[211], A. Howard[211], G. Iles[211], M. Knight[211], J. Langford[211], J. León Holgado[211], L. Lyons[211], A.-M. Magnan[211], S. Malik[211], M. Mieskolainen[211], J. Nash[211,213], M. Pesaresi[211], B. C. Radburn-Smith[211], A. Richards[211], A. Rose[211], K. Savva[211], C. Seez[211], R. Shukla[211], A. Tapper[211], K. Uchida[211], G. P. Uttley[211], L. H. Vage[211], T. Virdee[68,211], M. Vojinovic[211], N. Wardle[211], D. Winterbottom[211], K. Coldham[214], J. E. Cole[214], A. Khan[214], P. Kyberd[214], I. D. Reid[214], S. Abdullin[215], A. Brinkerhoff[215], B. Caraway[215], E. Collins[215], J. Dittmann[215], K. Hatakeyama[215], J. Hiltbrand[215], B. McMaster[215], M. Saunders[215], S. Sawant[215], C. Sutantawibul[215], J. Wilson[215], R. Bartek[216], A. Dominguez[216], C. Huerta Escamilla[216], A. E. Simsek[216], R. Uniyal[216], A. M. Vargas Hernandez[216], B. Bam[217], R. Chudasama[217], S. I. Cooper[217], S. V. Gleyzer[217], C. U. Perez[217], P. Rumerio[137,217], E. Usai[217], R. Yi[217], A. Akpinar[218], D. Arcaro[218], C. Cosby[218], Z. Demiragli[218], C. Erice[218], C. Fangmeier[218], C. Fernandez Madrazo[218], E. Fontanesi[218], D. Gastler[218], F. Golf[218], S. Jeon[218], I. Reed[218], J. Rohlf[218], K. Salyer[218], D. Sperka[218], D. Spitzbart[218], I. Suarez[218], A. Tsatsos[218], S. Yuan[218], A. G. Zecchinelli[218], G. Benelli[219], X. Coubez[57,219], D. Cutts[219], M. Hadley[219], U. Heintz[219], J. M. Hogan[219,220], T. Kwon[219], G. Landsberg[219], K. T. Lau[219], D. Li[219], J. Luo[219], S. Mondal[219], M. Narain[219,277], N. Pervan[219], S. Sagir[219,221], F. Simpson[219], M. Stamenkovic[219], X. Yan[219], W. Zhang[219], S. Abbott[222], J. Bonilla[222], C. Brainerd[222], R. Breedon[222], H. Cai[222], M. Calderon De La Barca Sanchez[222], M. Chertok[222], M. Citron[222], J. Conway[222], P. T. Cox[222], R. Erbacher[222], F. Jensen[222], O. Kukral[222], G. Mocellin[222], M. Mulhearn[222], D. Pellett[222], W. Wei[222], Y. Yao[222], F. Zhang[222], M. Bachtis[223], R. Cousins[223], A. Datta[223], G. Flores Avila[223], J. Hauser[223], M. Ignatenko[223], M. A. Iqbal[223], T. Lam[223], E. Manca[223], A. Nunez Del Prado[223], D. Saltzberg[223], V. Valuev[223], R. Clare[224], J. W. Gary[224], M. Gordon[224], G. Hanson[224], W. Si[224], S. Wimpenny[224,278], J. G. Branson[225], S. Cittolin[225], S. Cooperstein[225], D. Diaz[225], J. Duarte[225], L. Giannini[225], J. Guiang[225], R. Kansal[225], V. Krutelyov[225], R. Lee[225], J. Letts[225], M. Masciovecchio[225], F. Mokhtar[225], S. Mukherjee[225], M. Pieri[225], M. Quinnan[225], B. V. Sathia Narayanan[225], V. Sharma[225], M. Tadel[225], E. Vourliotis[225], F. Würthwein[225], Y. Xiang[225], A. Yagil[225], A. Barzdukas[226], L. Brennan[226], C. Campagnari[226], J. Incandela[226], J. Kim[226], A. J. Li[226], P. Masterson[226], H. Mei[226], J. Richman[226], U. Sarica[226], R. Schmitz[226], F. Setti[226], J. Sheplock[226], D. Stuart[226], T. Á. Vámi[226], S. Wang[226], A. Bornheim[227], O. Cerri[227], A. Latorre[227], J. Mao[227], H. B. Newman[227], M. Spiropulu[227], J. R. Vlimant[227], C. Wang[227], S. Xie[227], R. Y. Zhu[227], J. Alison[228], S. An[228], M. B. Andrews[228], P. Bryant[228], M. Cremonesi[228], V. Dutta[228], T. Ferguson[228], A. Harilal[228], C. Liu[228], T. Mudholkar[228], S. Murthy[228], P. Palit[228], M. Paulini[228], A. Roberts[228], A. Sanchez[228], W. Terrill[228], J. P. Cumalat[229], W. T. Ford[229], A. Hart[229], A. Hassani[229], G. Karathanasis[229], E. MacDonald[229], N. Manganelli[229], A. Perloff[229], C. Savard[229], N. Schonbeck[229], K. Stenson[229], K. A. Ulmer[229],

S. R. Wagner [229], N. Zipper [229], J. Alexander [230], S. Bright-Thonney [230], X. Chen [230], D. J. Cranshaw [230], J. Fan [230], X. Fan [230], D. Gadkari [230], S. Hogan [230], P. Kotamnives[230], J. Monroy [230], M. Oshiro [230], J. R. Patterson [230], J. Reichert [230], M. Reid [230], A. Ryd [230], J. Thom [230], P. Wittich [230], R. Zou [230], M. Albrow [122], M. Alyari [122], O. Amram [122], G. Apollinari [122], A. Apresyan [122], L. A. T. Bauerdick [122], D. Berry [122], J. Berryhill [122], P. C. Bhat [122], K. Burkett [122], J. N. Butler [122], A. Canepa [122], G. B. Cerati [122], H. W. K. Cheung [122], F. Chlebana [122], G. Cummings [122], J. Dickinson [122], I. Dutta [122], V. D. Elvira [122], Y. Feng [122], J. Freeman [122], A. Gandrakota [122], Z. Gecse [122], L. Gray [122], D. Green[122], A. Grummer [122], S. Grünendahl [122], D. Guerrero [122], O. Gutsche [122], R. M. Harris [122], R. Heller [122], T. C. Herwig [122], J. Hirschauer [122], L. Horyn [122], B. Jayatilaka [122], S. Jindariani [122], M. Johnson [122], U. Joshi [122], T. Klijnsma [122], B. Klima [122], K. H. M. Kwok [122], S. Lammel [122], D. Lincoln [122], R. Lipton [122], T. Liu [122], C. Madrid [122], K. Maeshima [122], C. Mantilla [122], D. Mason [122], P. McBride [122], P. Merkel [122], S. Mrenna [122], S. Nahn [122], J. Ngadiuba [122], D. Noonan [122], V. Papadimitriou [122], N. Pastika [122], K. Pedro [122], C. Pena [122,227], F. Ravera [122], A. Reinsvold Hall [122,231], L. Ristori [122], E. Sexton-Kennedy [122], N. Smith [122], A. Soha [122], L. Spiegel [122], S. Stoynev [122], J. Strait [122], L. Taylor [122], S. Tkaczyk [122], N. V. Tran [122], L. Uplegger [122], E. W. Vaandering [122], A. Whitbeck [122], I. Zoi [122], C. Aruta [232], P. Avery [232], D. Bourilkov [232], L. Cadamuro [232], P. Chang [232], V. Cherepanov [232], R. D. Field[232], E. Koenig [232], M. Kolosova [232], J. Konigsberg [232], A. Korytov [232], K. Matchev [232], N. Menendez [232], G. Mitselmakher [232], K. Mohrman [232], A. Muthirakalayil Madhu [232], N. Rawal [232], D. Rosenzweig [232], S. Rosenzweig [232], J. Wang [232], T. Adams [233], A. Al Kadhim [233], A. Askew [233], S. Bower [233], R. Habibullah [233], V. Hagopian [233], R. Hashmi [233], R. S. Kim [233], S. Kim [233], T. Kolberg [233], G. Martinez[233], H. Prosper [233], P. R. Prova[233], M. Wulansatiti [233], R. Yohay [233], J. Zhang[233], B. Alsufyani[234], M. M. Baarmand [234], S. Butalla [234], T. Elkafrawy [234,274], M. Hohlmann [234], R. Kumar Verma [234], M. Rahmani[234], E. Yanes[234], M. R. Adams [235], A. Baty [235], C. Bennett[235], R. Cavanaugh [235], R. Escobar Franco [235], O. Evdokimov [235], C. E. Gerber [235], D. J. Hofman [235], J. H. Lee [235], D. S. Lemos [235], A. H. Merrit [235], C. Mills [235], S. Nanda [235], G. Oh [235], B. Ozek [235], D. Pilipovic [235], R. Pradhan [235], T. Roy [235], S. Rudrabhatla [235], M. B. Tonjes [235], N. Varelas [235], Z. Ye [235], J. Yoo [235], M. Alhusseini [236], D. Blend[236], K. Dilsiz [236,237], L. Emediato [236], G. Karaman [236], O. K. Köseyan [236], J.-P. Merlo[236], A. Mestvirishvili [55,236], J. Nachtman [236], O. Neogi[236], H. Ogul [236,238], Y. Onel [236], A. Penzo [236], C. Snyder[236], E. Tiras [236,239], B. Blumenfeld [240], L. Corcodilos [240], J. Davis [240], A. V. Gritsan [240], L. Kang [240], S. Kyriacou [240], P. Maksimovic [240], M. Roguljic [240], J. Roskes [240], S. Sekhar [240], M. Swartz [240], A. Abreu [241], L. F. Alcerro Alcerro [241], J. Anguiano [241], P. Baringer [241], A. Bean [241], Z. Flowers [241], D. Grove [241], J. King [241], G. Krintiras [241], M. Lazarovits [241], C. Le Mahieu [241], J. Marquez [241], N. Minafra [241], M. Murray [241], M. Nickel [241], M. Pitt [241], S. Popescu [241,242], C. Rogan [241], C. Royon [241], R. Salvatico [241], S. Sanders [241], C. Smith [241], Q. Wang [241], G. Wilson [241], B. Allmond [243], A. Ivanov [243], K. Kaadze [243], A. Kalogeropoulos [243], D. Kim[243], Y. Maravin [243], J. Natoli [243], D. Roy [243], G. Sorrentino [243], F. Rebassoo [244], D. Wright [244], A. Baden [245], A. Belloni [245], Y. M. Chen [245], S. C. Eno [245], N. J. Hadley [245], S. Jabeen [245], R. G. Kellogg [245], T. Koeth [245], Y. Lai [245], S. Lascio [245], A. C. Mignerey [245], S. Nabili [245], C. Palmer [245], C. Papageorgakis [245], M. M. Paranjpe[245], L. Wang [245], J. Bendavid [246], I. A. Cali [246], M. D'Alfonso [246], J. Eysermans [246], C. Freer [246], G. Gomez-Ceballos [246], M. Goncharov[246], G. Grosso[246], P. Harris[246], D. Hoang[246], D. Kovalskyi [246], J. Krupa [246], L. Lavezzo [246], Y.-J. Lee [246], K. Long [246], A. Novak [246], C. Paus [246], D. Rankin [246], C. Roland [246], G. Roland [246], S. Rothman [246], G. S. F. Stephans [246], Z. Wang [246], B. Wyslouch [246], T. J. Yang [246], B. Crossman [247], B. M. Joshi [247], C. Kapsiak [247], M. Krohn [247], D. Mahon [247], J. Mans [247], B. Marzocchi [247], S. Pandey [247], M. Revering [247], R. Rusack [247], R. Saradhy [247], N. Schroeder [247], N. Strobbe [247], M. A. Wadud [247], L. M. Cremaldi [248], K. Bloom [249], D. R. Claes [249], G. Haza [249], J. Hossain [249], C. Joo [249], I. Kravchenko [249], J. E. Siado [249], W. Tabb [249], A. Vagnerini [249], A. Wightman [249], F. Yan [249], D. Yu [249], H. Bandyopadhyay [250], L. Hay [250], I. Iashvili [250], A. Kharchilava [250], M. Morris [250], D. Nguyen [250], S. Rappoccio [250], H. Rejeb Sfar[250], A. Williams [250], G. Alverson [251], E. Barberis [251], J. Dervan[251], Y. Haddad [251], Y. Han [251], A. Krishna [251], J. Li [251], M. Lu [251], G. Madigan [251], R. Mccarthy [251], D. M. Morse [251], V. Nguyen [251], T. Orimoto [251], A. Parker [251], L. Skinnari [251], B. Wang [251], D. Wood [251], S. Bhattacharya [252], J. Bueghly[252], Z. Chen [252], S. Dittmer [252], K. A. Hahn [252], Y. Liu [252], Y. Miao [252], D. G. Monk [252], M. H. Schmitt [252], A. Taliercio [252], M. Velasco[252], G. Agarwal [253], R. Band [253], R. Bucci[253], S. Castells [253], A. Das [253], R. Goldouzian [253], M. Hildreth [253], K. W. Ho [253], K. Hurtado Anampa [253], T. Ivanov [253], C. Jessop [253], K. Lannon [253], J. Lawrence [253], N. Loukas [253], L. Lutton [253], J. Mariano[253], N. Marinelli[253], I. Mcalister[253], T. McCauley [253], C. Mcgrady [253], C. Moore [253], Y. Musienko [253,280], H. Nelson [253], M. Osherson [253], A. Piccinelli [253], R. Ruchti [253], A. Townsend [253], Y. Wan[253], M. Wayne [253], H. Yockey[253], M. Zarucki [253], L. Zygala [253], A. Basnet [254], B. Bylsma[254], M. Carrigan [254], L. S. Durkin [254], C. Hill [254], M. Joyce [254], M. Nunez Ornelas [254], K. Wei[254], B. L. Winer [254], B. R. Yates [254],

F. M. Addesa [255], H. Bouchamaoui [255], P. Das [255], G. Dezoort [255], P. Elmer [255], A. Frankenthal [255], B. Greenberg [255], N. Haubrich [255], G. Kopp [255], S. Kwan [255], D. Lange [255], A. Loeliger [255], D. Marlow [255], I. Ojalvo [255], J. Olsen [255], A. Shevelev [255], D. Stickland [255], C. Tully [255], S. Malik [256], A. S. Bakshi [50], V. E. Barnes [50], S. Chandra [50], R. Chawla [50], S. Das [50], A. Gu [50], L. Gutay [50], M. Jones [50], A. W. Jung [50], D. Kondratyev [50], A. M. Koshy [50], M. Liu [50], G. Negro [50], N. Neumeister [50], G. Paspalaki [50], S. Piperov [50], V. Scheurer [50], J. F. Schulte [50], M. Stojanovic [50], J. Thieman [50], A. K. Virdi [50], F. Wang [50], W. Xie [50], J. Dolen [257], N. Parashar [257], A. Pathak [257], D. Acosta [258], T. Carnahan [258], K. M. Ecklund [258], P. J. Fernández Manteca [258], S. Freed [258], P. Gardner [258], F. J. M. Geurts [258], W. Li [258], O. Miguel Colin [258], B. P. Padley [258], R. Redjimi [258], J. Rotter [258], E. Yigitbasi [258], Y. Zhang [258], A. Bodek [259], P. de Barbaro [259], R. Demina [259], J. L. Dulemba [259], A. Garcia-Bellido [259], O. Hindrichs [259], A. Khukhunaishvili [259], N. Parmar [259], P. Parygin [259,281], E. Popova [259,281], R. Taus [259], K. Goulianos [260], B. Chiarito [261], J. P. Chou [261], S. V. Clark [261], Y. Gershtein [261], E. Halkiadakis [261], M. Heindl [261], C. Houghton [261], D. Jaroslawski [261], O. Karacheban [65,261], I. Laflotte [261], A. Lath [261], R. Montalvo [261], K. Nash [261], H. Routray [261], S. Salur [261], S. Schnetzer [261], S. Somalwar [261], R. Stone [261], S. A. Thayil [261], S. Thomas [261], J. Vora [261], H. Wang [261], H. Acharya [262], D. Ally [262], A. G. Delannoy [262], S. Fiorendi [262], S. Higginbotham [262], T. Holmes [262], A. R. Kanuganti [262], N. Karunarathna [262], L. Lee [262], E. Nibigira [262], S. Spanier [262], D. Aebi [263], M. Ahmad [263], O. Bouhali [263,264], R. Eusebi [263], J. Gilmore [263], T. Huang [263], T. Kamon [141,263], H. Kim [263], S. Luo [263], R. Mueller [263], D. Overton [263], D. Rathjens [263], A. Safonov [263], N. Akchurin [265], J. Damgov [265], V. Hegde [265], A. Hussain [265], Y. Kazhykarim [265], K. Lamichhane [265], S. W. Lee [265], A. Mankel [265], T. Peltola [265], I. Volobouev [265], E. Appelt [266], Y. Chen [266], S. Greene [266], A. Gurrola [266], W. Johns [266], R. Kunnawalkam Elayavalli [266], A. Melo [266], F. Romeo [266], P. Sheldon [266], S. Tuo [266], J. Velkovska [266], J. Viinikainen [266], B. Cardwell [267], B. Cox [267], J. Hakala [267], R. Hirosky [267], A. Ledovskoy [267], C. Neu [267], C. E. Perez Lara [267], P. E. Karchin [268], A. Aravind [269], S. Banerjee [269], K. Black [269], T. Bose [269], S. Dasu [269], I. De Bruyn [269], P. Everaerts [269], C. Galloni [269], H. He [269], M. Herndon [269], A. Herve [269], C. K. Koraka [269], A. Lanaro [269], R. Loveless [269], J. Madhusudanan Sreekala [269], A. Mallampalli [269], A. Mohammadi [269], S. Mondal [269], G. Parida [269], L. Pétré [269], D. Pinna [269], A. Savin [269], V. Shang [269], V. Sharma [269], W. H. Smith [269], D. Teague [269], H. F. Tsoi [269], W. Vetens [269], A. Warden [269], S. Afanasiev [282], V. Andreev [282], Yu. Andreev [282], T. Aushev [282], M. Azarkin [282], A. Babaev [282], A. Belyaev [282], V. Blinov [282,283], E. Boos [282], V. Borshch [282], D. Budkouski [282], V. Chekhovsky [282], R. Chistov [282,283], M. Danilov [282,283], A. Dermenev [282], T. Dimova [282,283], D. Druzhkin [5,282], A. Ershov [282], G. Gavrilov [282], V. Gavrilov [282], S. Gninenko [282], V. Golovtsov [282], N. Golubev [282], I. Golutvin [282], I. Gorbunov [282], A. Gribushin [282], Y. Ivanov [282], V. Kachanov [282], A. Kaminskiy [282], V. Karjavine [282], A. Karneyeu [282], L. Khein [282], V. Kim [282,283], M. Kirakosyan [282], D. Kirpichnikov [282], M. Kirsanov [282], O. Kodolova [1,282], D. Konstantinov [282], V. Korenkov [282], V. Korotkikh [282], A. Kozyrev [282,283], N. Krasnikov [282], A. Lanev [282], P. Levchenko [251,282], N. Lychkovskaya [282], V. Makarenko [282], A. Malakhov [282], V. Matveev [282,283], V. Murzin [282], A. Nikitenko [1,211,282], S. Obraztsov [282], V. Oreshkin [282], V. Palichik [282], V. Perelygin [282], S. Petrushanko [282], S. Polikarpov [282,283], V. Popov [282], O. Radchenko [282,283], M. Savina [282], V. Savrin [282], V. Shalaev [282], S. Shmatov [282], S. Shulha [282], Y. Skovpen [282,283], S. Slabospitskii [282], V. Smirnov [282], A. Snigirev [282], D. Sosnov [282], V. Sulimov [282], E. Tcherniaev [282], A. Terkulov [282], O. Teryaev [282], I. Tlisova [282], A. Toropin [282], L. Uvarov [282], A. Uzunian [282], I. Vardanyan [282], A. Vorobyev [279,282], N. Voytishin [282], B. S. Yuldashev [270,282], A. Zarubin [282], I. Zhizhin [282] & A. Zhokin [282]

[1]Yerevan Physics Institute, Yerevan, Armenia. [2]Yerevan State University, Yerevan, Armenia. [3]Institut für Hochenergiephysik, Vienna, Austria. [4]TU Wien, Vienna, Austria. [5]Universiteit Antwerpen, Antwerpen, Belgium. [6]Institute of Basic and Applied Sciences, Faculty of Engineering, Arab Academy for Science, Technology and Maritime Transport, Alexandria, Egypt. [7]Vrije Universiteit Brussel, Brussel, Belgium. [8]Ghent University, Ghent, Belgium. [9]Université Libre de Bruxelles, Bruxelles, Belgium. [10]Université Catholique de Louvain, Louvain-la-Neuve, Belgium. [11]Centro Brasileiro de Pesquisas Fisicas, Rio de Janeiro, Brazil. [12]Universidade do Estado do Rio de Janeiro, Rio de Janeiro, Brazil. [13]Universidade Estadual de Campinas, Campinas, Brazil. [14]Federal University of Rio Grande do Sul, Porto Alegre, Brazil. [15]UFMS, Nova Andradina, Brazil. [16]Universidade Estadual Paulista, Universidade Federal do ABC, São Paulo, Brazil. [17]Institute for Nuclear Research and Nuclear Energy, Bulgarian Academy of Sciences, Sofia, Bulgaria. [18]University of Sofia, Sofia, Bulgaria. [19]Instituto De Alta Investigación, Universidad de Tarapacá, Arica, Chile. [20]Beihang University, Beijing, China. [21]Department of Physics, Tsinghua University, Beijing, China. [22]Nanjing Normal University, Nanjing, China. [23]Institute of High Energy Physics, Beijing, China. [24]University of Chinese Academy of Sciences, Beijing, China. [25]China Center of Advanced Science and Technology, Beijing, China. [26]China Spallation Neutron Source, Guangdong, China. [27]State Key Laboratory of Nuclear Physics and Technology, Peking University, Beijing, China. [28]Sun Yat-Sen University, Guangzhou, China. [29]University of Science and Technology of China, Hefei, China. [30]Institute of Modern Physics and Key Laboratory of Nuclear Physics and Ion-beam Application (MOE) - Fudan University, Shanghai, China. [31]Zhejiang University, Hangzhou, Zhejiang, China. [32]Universidad de Los Andes, Bogota, Colombia. [33]Universidad de Antioquia, Medellin, Colombia. [34]University of Split, Faculty of Electrical Engineering, Mechanical Engineering and Naval Architecture, Split, Croatia. [35]University of Split, Faculty of Science, Split, Croatia. [36]Institute Rudjer Boskovic, Zagreb, Croatia. [37]University of Cyprus, Nicosia, Cyprus. [38]Charles University, Prague, Czech Republic. [39]Escuela Politecnica Nacional, Quito, Ecuador. [40]Universidad San Francisco de Quito, Quito, Ecuador. [41]Academy of Scientific Research and Technology of the Arab Republic of Egypt, Egyptian Network of High Energy Physics, Cairo, Egypt. [42]Helwan University, Cairo, Egypt. [43]British University in Egypt, Cairo, Egypt. [44]Center for High

Energy Physics (CHEP-FU), Fayoum University, El-Fayoum, Egypt. [45]National Institute of Chemical Physics and Biophysics, Tallinn, Estonia. [46]Department of Physics, University of Helsinki, Helsinki, Finland. [47]Helsinki Institute of Physics, Helsinki, Finland. [48]Lappeenranta-Lahti University of Technology, Lappeenranta, Finland. [49]IRFU, CEA, Université Paris-Saclay, Gif-sur-Yvette, France. [50]Purdue University, West Lafayette, Indiana, USA. [51]Laboratoire Leprince-Ringuet, CNRS/IN2P3, Ecole Polytechnique, Institut Polytechnique de Paris, Palaiseau, France. [52]Université de Strasbourg CNRS, Strasbourg, France. [53]Université de Haute Alsace, Mulhouse, France. [54]Institut de Physique des 2 Infinis de Lyon (IP2I), Villeurbanne, France. [55]Georgian Technical University, Tbilisi, Georgia. [56]RWTH Aachen University I. Physikalisches Institut, Aachen, Germany. [57]RWTH Aachen University III. Physikalisches Institut A, Aachen, Germany. [58]The University of the State of Amazonas, Manaus, Brazil. [59]RWTH Aachen University III. Physikalisches Institut B, Aachen, Germany. [60]Erzincan Binali Yildirim University, Erzincan, Turkey. [61]Deutsches Elektronen-Synchrotron, Hamburg, Germany. [62]University of Hamburg, Hamburg, Germany. [63]Isfahan University of Technology, Isfahan, Iran. [64]Bergische UniversityWuppertal (BUW), Wuppertal, Germany. [65]Brandenburg University of Technology, Cottbus, Germany. [66]Forschungszentrum Jülich, Juelich, Germany. [67]Karlsruher Institut fuer Technologie, Karlsruhe, Germany. [68]CERN European Organization for Nuclear Research, Geneva, Switzerland. [69]Institute of Nuclear and Particle Physics (INPP), NCSR Demokritos, Aghia Paraskevi, Greece. [70]National and Kapodistrian University of Athens, Athens, Greece. [71]National Technical University of Athens, Athens, Greece. [72]University of Ioánnina, Ioánnina, Greece. [73]HUN-RENWigner Research Centre for Physics, Budapest, Hungary. [74]Institute of Physics, University of Debrecen, Debrecen, Hungary. [75]Institute of Nuclear Research ATOMKI, Debrecen, Hungary. [76]MTA-ELTE Lendület CMS Particle and Nuclear Physics Group, Eötvös Loránd University, Budapest, Hungary. [77]Physics Department, Faculty of Science, Assiut University, Assiut, Egypt. [78]Faculty of Informatics, University of Debrecen, Debrecen, Hungary. [79]Karoly Robert Campus, MATE Institute of Technology, Gyongyos, Hungary. [80]Panjab University, Chandigarh, India. [81]Punjab Agricultural University, Ludhiana, India. [82]University of Delhi, Delhi, India. [83]Saha Institute of Nuclear Physics HBNI, Kolkata, India. [84]University of Visva-Bharati, Santiniketan, India. [85]Indian Institute of Technology Madras, Madras, India. [86]Indian Institute of Science (IISc), Bangalore, India. [87]Tata Institute of Fundamental Research-A, Mumbai, India. [88]Tata Institute of Fundamental Research-B, Mumbai, India. [89]Birla Institute of Technology Mesra, Mesra, India. [90]National Institute of Science Education and Research, An OCC of Homi Bhabha National Institute, Bhubaneswar, Odisha, India. [91]IIT Bhubaneswar, Bhubaneswar, India. [92]Institute of Physics, Bhubaneswar, India. [93]Indian Institute of Science Education and Research (IISER), Pune, India. [94]University of Hyderabad, Hyderabad, India. [95]Department of Physics, Isfahan University of Technology, Isfahan, Iran. [96]Sharif University of Technology, Tehran, Iran. [97]Institute for Research in Fundamental Sciences (IPM), Tehran, Iran. [98]Department of Physics, University of Science and Technology of Mazandaran, Behshahr, Iran. [99]University College Dublin, Dublin, Ireland. [100]INFN Sezione di Bari, Bari, Italy. [101]Università di Bari, Bari, Italy. [102]Politecnico di Bari, Bari, Italy. [103]INFN Sezione di Bologna, Bologna, Italy. [104]Università di Bologna, Bologna, Italy. [105]Italian National Agency for New Technologies, Energy and Sustainable Economic Development, Bologna, Italy. [106]INFN Sezione di Catania, Catania, Italy. [107]Università di Catania, Catania, Italy. [108]Centro Siciliano di Fisica Nucleare e di Struttura Della Materia, Catania, Italy. [109]INFN Sezione di Firenze, Firenze, Italy. [110]Università di Firenze, Firenze, Italy. [111]INFN Laboratori Nazionali di Frascati, Frascati, Italy. [112]Università degli Studi Guglielmo Marconi, Roma, Italy. [113]INFN Sezione di Genova, Genova, Italy. [114]Università di Genova, Genova, Italy. [115]INFN Sezione di Milano-Bicocca, Milano, Italy. [116]Università di Milano-Bicocca, Milano, Italy. [117]INFN Sezione di Napoli, Napoli, Italy. [118]Università di Napoli 'Federico II', Napoli, Italy. [119]Università della Basilicata, Potenza, Italy. [120]Scuola Superiore Meridionale, Università di Napoli 'Federico II', Napoli, Italy. [121]INFN Sezione di Padova, Padova, Italy. [122]Fermi National Accelerator Laboratory, Batavia, IL, USA. [123]Laboratori Nazionali di Legnaro dell'INFN, Legnaro, Italy. [124]Università di Padova, Padova, Italy. [125]INFN Sezione di Pavia, Pavia, Italy. [126]Università di Pavia, Pavia, Italy. [127]INFN Sezione di Perugia, Perugia, Italy. [128]Università di Perugia, Perugia, Italy. [129]Consiglio Nazionale delle Ricerche—Istituto Officina dei Materiali, Perugia, Italy. [130]INFN Sezione di Pisa, Pisa, Italy. [131]Università di Pisa, Pisa, Italy. [132]Scuola Normale Superiore di Pisa, Pisa, Italy. [133]Università di Siena, Siena, Italy. [134]INFN Sezione di Roma, Roma, Italy. [135]Sapienza Università di Roma, Roma, Italy. [136]INFN Sezione di Torino, Torino, Italy. [137]Università di Torino, Torino, Italy. [138]Università del Piemonte Orientale, Novara, Italy. [139]INFN Sezione di Trieste, Trieste, Italy. [140]Università di Trieste, Trieste, Italy. [141]Kyungpook National University, Daegu, Korea. [142]Department of Mathematics and Physics - GWNU, Gangneung, Korea. [143]Chonnam National University, Institute for Universe and Elementary Particles, Kwangju, Korea. [144]Hanyang University, Seoul, Korea. [145]Korea University, Seoul, Korea. [146]Department of Physics, Kyung Hee University, Seoul, Korea. [147]Sejong University, Seoul, Korea. [148]Seoul National University, Seoul, Korea. [149]University of Seoul, Seoul, Korea. [150]Department of Physics, Yonsei University, Seoul, Korea. [151]Sungkyunkwan University, Suwon, Korea. [152]College of Engineering and Technology, American University of the Middle East (AUM), Dasman, Kuwait. [153]Riga Technical University, Riga, Latvia. [154]University of Latvia (LU), Riga, Latvia. [155]Vilnius University, Vilnius, Lithuania. [156]National Centre for Particle Physics, Universiti Malaya, Kuala Lumpur, Malaysia. [157]Department of Applied Physics, Faculty of Science and Technology, Universiti Kebangsaan Malaysia, Bangi, Malaysia. [158]Universidad de Sonora (UNISON), Hermosillo, Mexico. [159]Centro de Investigacion y de Estudios Avanzados del IPN, Mexico City, Mexico. [160]Consejo Nacional de Ciencia y Tecnología, Mexico City, Mexico. [161]Universidad Iberoamericana, Mexico City, Mexico. [162]Benemerita Universidad Autonoma de Puebla, Puebla, Mexico. [163]University of Montenegro, Podgorica, Montenegro. [164]University of Canterbury, Christchurch, New Zealand. [165]National Centre for Physics, Quaid-I-Azam University, Islamabad, Pakistan. [166]AGH University of Krakow, Faculty of Computer Science Electronics and Telecommunications, Krakow, Poland. [167]National Centre for Nuclear Research, Swierk, Poland. [168]Institute of Experimental Physics, Faculty of Physics, University of Warsaw, Warsaw, Poland. [169]Warsaw University of Technology, Warsaw, Poland. [170]Laboratório de Instrumenta, cão e Física Experimental de Partículas, Lisboa, Portugal. [171]Faculty of Physics, University of Belgrade, Belgrade, Serbia. [172]VINCA Institute of Nuclear Sciences, University of Belgrade, Belgrade, Serbia. [173]Centro de Investigaciones Energéticas Medioambientales y Tecnológicas (CIEMAT), Madrid, Spain. [174]Universidad Autónoma de Madrid, Madrid, Spain. [175]Universidad de Oviedo, Instituto Universitario de Ciencias y Tecnologías Espaciales de Asturias (ICTEA), Oviedo, Spain. [176]Instituto de Física de Cantabria (IFCA), CSIC-Universidad de Cantabria, Santander, Spain. [177]University of Colombo, Colombo, Sri Lanka. [178]Trincomalee Campus, Eastern University Sri Lanka, Nilaveli, Sri Lanka. [179]Department of Physics, University of Ruhuna, Matara, Sri Lanka. [180]Saegis Campus, Nugegoda, Sri Lanka. [181]Ecole Polytechnique Fédérale Lausanne, Lausanne, Switzerland. [182]Paul Scherrer Institut, Villigen, Switzerland. [183]Universität Zürich, Zurich, Switzerland. [184]ETH Zurich—Institute for Particle Physics and Astrophysics (IPA), Zurich, Switzerland. [185]Stefan Meyer Institute for Subatomic Physics, Vienna, Austria. [186]National Central University, Chung-Li, Taiwan. [187]Laboratoire d'Annecy-le-Vieux de Physique des Particules IN2P3-CNRS, Annecy-le-Vieux, France. [188]National Taiwan University (NTU), Taipei, Taiwan. [189]High Energy Physics Research Unit, Department of Physics, Faculty of Science, Chulalongkorn University, Bangkok, Thailand. [190]Physics Department Science and Art Faculty, Çukurova University, Adana, Turkey. [191]Near East University, Research Center of Experimental Health Science, Mersin, Turkey. [192]Konya Technical University, Konya, Turkey. [193]Izmir Bakircay University, Izmir, Turkey. [194]Adiyaman University, Adiyaman, Turkey. [195]Physics Department Middle East Technical University, Physics Department, Ankara, Turkey. [196]Bozok Universitetesi Rektörlügü, Yozgat, Turkey. [197]Bogazici University, Istanbul, Turkey. [198]Marmara University, Istanbul, Turkey. [199]Milli Savunma University, Istanbul, Turkey. [200]Kafkas University, Kars, Turkey. [201]Istanbul Technical University, Istanbul, Turkey. [202]Hacettepe University, Ankara, Turkey. [203]Istanbul University, Istanbul, Turkey. [204]Faculty of Engineering, Istanbul University—Cerrahpasa, Istanbul, Turkey. [205]Yildiz Technical University, Istanbul, Turkey. [206]Institute for Scintillation Materials of National Academy of Science of Ukraine, Kharkiv, Ukraine. [207]National Science Centre, Kharkiv Institute of Physics and Technology, Kharkiv, Ukraine. [208]University of Bristol, Bristol, UK. [209]Rutherford Appleton Laboratory, Didcot, UK. [210]School of Physics and Astronomy, University of Southampton, Southampton, UK. [211]Imperial College, London, UK. [212]IPPP

Durham University, Durham, UK. [213]Faculty of Science, Monash University, Clayton, Australia. [214]Brunel University, Uxbridge, UK. [215]Baylor University, Waco, TX, USA. [216]Catholic University of America, Washington, DC, USA. [217]The University of Alabama, Tuscaloosa, AL, USA. [218]Boston University, Boston, MA, USA. [219]Brown University, Providence, RI, USA. [220]Bethel University, St. Paul, MN, USA. [221]Karamanoğlu Mehmetbey University, Karaman, Turkey. [222]University of California Davis, Davis, CA, USA. [223]University of California, Los Angeles, CA, USA. [224]University of California Riverside, Riverside, CA, USA. [225]University of California San Diego, La Jolla, CA, USA. [226]Department of Physics, University of CaliforniaSanta Barbara, Santa Barbara, CA, USA. [227]California Institute of Technology, Pasadena, CA, USA. [228]Carnegie Mellon University, Pittsburgh, PA, USA. [229]University of Colorado Boulder, Boulder, CO, USA. [230]Cornell University, Ithaca, NY, USA. [231]United States Naval Academy, Annapolis, MD, USA. [232]University of Florida, Gainesville, FL, USA. [233]Florida State University, Tallahassee, FL, USA. [234]Florida Institute of Technology, Melbourne, FL, USA. [235]University of Illinois Chicago, Chicago USA, Chicago, USA. [236]The University of Iowa, Iowa City, IA, USA. [237]Bingol University, Bingol, Turkey. [238]Sinop University, Sinop, Turkey. [239]Erciyes University, Kayseri, Turkey. [240]Johns Hopkins University, Baltimore, MD, USA. [241]The University of Kansas, Lawrence, KS, USA. [242]Horia Hulubei National Institute of Physics and Nuclear Engineering (IFIN-HH), Bucharest, Romania. [243]Kansas State University, Manhattan, KS, USA. [244]Lawrence Livermore National Laboratory, Livermore, CA, USA. [245]University of Maryland, College Park, MD, USA. [246]Massachusetts Institute of Technology, Cambridge, MA, USA. [247]University of Minnesota, Minneapolis, MN, USA. [248]University of Mississippi, Oxford, MS, USA. [249]University of Nebraska-Lincoln, Lincoln, NE, USA. [250]State University of New York at Buffalo, Buffalo, NY, USA. [251]Northeastern University, Boston, MA, USA. [252]Northwestern University, Evanston, IL, USA. [253]University of Notre Dame, Notre Dame, IN, USA. [254]The Ohio State University, Columbus, OH, USA. [255]Princeton University, Princeton, NJ, USA. [256]University of Puerto Rico, Mayaguez, PR, USA. [257]Purdue University Northwest, Hammond, IN, USA. [258]Rice University, Houston, TX, USA. [259]University of Rochester, Rochester, NY, USA. [260]The Rockefeller University, New York, NY, USA. [261]Rutgers The State University of New Jersey, Piscataway, NJ, USA. [262]University of Tennessee, Knoxville, TN, USA. [263]Texas A&M University, College Station, TX, USA. [264]Texas A&M University at Qatar, Doha, Qatar. [265]Texas Tech University, Lubbock, TX, USA. [266]Vanderbilt University, Nashville, TN, USA. [267]University of Virginia, Charlottesville, VI, USA. [268]Wayne State University, Detroit, MI, USA. [269]University of Wisconsin - Madison, Madison, WI, USA. [270]Institute of Nuclear Physics of the Uzbekistan Academy of Sciences, Tashkent, Uzbekistan. [271]Present address: The University of Iowa, Iowa City, IA, USA. [272]Present address: Henan Normal University, Xinxiang, China. [273]Present address: Zewail City of Science and Technology, Zewail, Egypt. [274]Present address: Ain Shams University, Cairo, Egypt. [275]Present address: Universitatea Babes-Bolyai - Facultatea de Fizica, Cluj-Napoca, Romania. [276]Present address: Stanbul Okan University, Istanbul, Turkey. [277]Deceased: M. Narain. [278]Deceased: S. Wimpenny. [279]Deceased: A. Vorobyev. [280]Also at an institute or an international laboratory covered by a cooperation agreement with CERN: A. Starodumov, Z. Tsamalaidze, Y. Musienko. [281]Now at an institute or an international laboratory covered by a cooperation agreement with CERN: P. Parygin, E. Popova. [282]Authors affiliated with an institute or an international laboratory covered by a cooperation agreement with CERN: S. Afanasiev, V. Andreev, Yu. Andreev, T. Aushev, M. Azarkin, A. Babaev, A. Belyaev, V. Blinov, E. Boos, V. Borshch, D. Budkouski, V. Chekhovsky, R. Chistov, M. Danilov, A. Dermenev, T. Dimova, D. Druzhkin, A. Ershov, G. Gavrilov, V. Gavrilov, S. Gninenko, V. Golovtcov, N. Golubev, I. Golutvin, I. Gorbunov, A. Gribushin, Y. Ivanov, V. Kachanov, A. Kaminskiy, V. Karjavine, A. Karneyeu, L. Khein, V. Kim, M. Kirakosyan, D. Kirpichnikov, M. Kirsanov, O. Kodolova, D. Konstantinov, V. Korenkov, V. Korotkikh, A. Kozyrev, N. Krasnikov, A. Lanev, P. Levchenko, N. Lychkovskaya, V. Makarenko, A. Malakhov, V. Matveev, V. Murzin, A. Nikitenko, S. Obraztsov, V. Oreshkin, V. Palichik, V. Perelygin, S. Petrushanko, S. Polikarpov, V. Popov, O. Radchenko, M. Savina, V. Savrin, V. Shalaev, S. Shmatov, S. Shulha, Y. Skovpen, S. Slabospitskii, V. Smirnov, A. Snigirev, D. Sosnov, V. Sulimov, E. Tcherniaev, A. Terkulov, O. Teryaev, I. Tlisova, A. Toropin, L. Uvarov, A. Uzunian, I. Vardanyan, A. Vorobyev, N. Voytishin, B. S. Yuldashev, A. Zarubin, I. Zhizhin, A. Zhokin. [283]Also at another institute or international laboratory covered by a cooperation agreement with CERN: V. Blinov, R. Chistov, M. Danilov, T. Dimova, V. Kim, A. Kozyrev, V. Matveev, S. Polikarpov, O. Radchenko, Y. Skovpen.

