## [Transparent Peer Review file · Nature Communications]

Elliptic anisotropy measurement of the $f_0(980)$ hadron in proton-lead collisions and evidence for its quark-antiquark composition

Corresponding Author: Dr Publication CMS_Collaboration

Version 0:

Reviewer comments:

Reviewer #1

(Remarks to the Author)

The authors have measured elliptic event anisotropy of f_0 in pPb collisions at LHC and used the number of constituent quark (NCQ) scaling of v_2 to test the quark content of the f_0 hadron assuming the quark coalescence mechanism for the hadron formation. The method is well established in high-energy heavy-ion physics field, while the NCQ scaling is approximately hold for various mesons and baryons. The presented results on f_0 v_2 measurements are interesting and worth publishing, therefore I would like to suggest this manuscript to be published, after a few more additional information presented in the manuscript or most likely in the method section as listed in the following.

1) Concerning on the invariant mass fitting, while the poorly described region below the mass peak of $\rho(770)$ might not be affecting too much on the final results of f_0 v_2 value, the v_2 values of the residual background as well as f_2 and $\rho(770)$, would need to be treated properly, so I would request authors to provide how other contributions behave especially the residual (and combinatorial) BG v_2 below the f_0 peak. This is especially because the residual BG is fitted by 3rd order polynomial function, I would like to know if this is totally independent polynomial function across the different azimuthal bins or if there is any constraint.

2) The analysis is limited within the multiplicity region in pPb collisions in order to be compared with other similar measurements on other light and strangeness hadrons, it is essential to provide the centrality (or multiplicity) dependence of v_2 measurements especially in the small system, since the non-flow subtraction is quite important in this case, therefore I would request authors to show some multiplicity dependence of this measurements in pPb collisions. It might be better to repeat the same measurements in pp collisions especially for the non-flow subtraction, but the lower multiplicity pPb collisions from the same dataset could be easier and good enough. I would like to see a kind of confirmation of the non-flow subtraction assumption taken from the K0S measurement.

Reviewer #2

(Remarks to the Author)

The authors of the manuscript use a novel method to estimate the quark structure of $f_0(980)$ meson. The method is based on previously published observations showing that elliptic flow (v_2) of mesons and baryons exhibits a number of constituent quarks (n_q) scaling. This empirically established scaling may be understood e.g. within phenomenological models of coalescence as a consequence of hadron formation which proceeds from quarks that are nearby in the phase space. This empirical scaling can then be compared between $f_0(980)$ and other hadrons, allowing to test the hypothesis that $f_0(980)$ contains 2 or 4 quarks ($n_q=2$ or $n_q=4$), which was done in the manuscript. Such a method was applied for the first time, and if established, it may become a valuable tool for understanding the structure of hadronic resonances. The results presented in the manuscript are certainly significant and novel, and they have the potential to significantly advance our understanding of the structure of specific hadron resonances. The topic of the structure of hadron resonances touches on the fundamental open questions connected with strong interaction. The results, therefore, deserve attention from a broader audience.

The experimental methods used in the paper are sound, and statistical analysis is precise. At the same time, I need to raise several important questions that have to be addressed before I can recommend the manuscript for publication in Nature Physics. These are the following:

- a) Equation (3). No reference to the theory paper is provided for Eq (3). How is the functional form of Eq (3) connected with the n_q scaling of v_2 discussed earlier in the manuscript (formula $v_n \sim n_q * v_n(p_T/n_q)$)? Reference [41], which is the main theoretical paper used here to advocate the scaling, does not give formula (3). Neither reference [39], which is the original theoretical proposal for this kind of analysis, does. The presence of the exponential term and " E_T " instead of p_T should be advocated by theory. Please clarify and provide the reference to relevant papers.
- b) Figure 3. Figure quotes "PRL 121 (2018) 082301" for datapoints other than those for $f_0(980)$. Datapoints in the PRL paper, however, do not contain datapoints for K_s^0 meson for $KE_T/n_q > \sim 2.5$ GeV shown here (compare the upper right panel of Fig.3 in the PRL paper with this Figure). How is that possible, please? Were they published in some other, unquoted paper? Or were they reconstructed within the presented analysis? If so, it is not documented in the manuscript. Can you please explain? Those datapoints are crucial for establishing the 7.7 significance shown here.
- c) Subsection A.2. Analysis works under the assumption that measured tracks are pions. PDG, however, says that the KK channel may constitute something between 20% to 50% of hadronical decay width. The $\pi^+\pi^-$ channel, therefore, does not constitute 100% of the hadronic decay width. How does the result change when taking this into account?
- d) Page 6, sentence starting "As shown in Fig.3, the NCQ scaling ...". This sentence and the following sentence speak about p_T/n_q while x-axis is KE_T/n_q . Is that a typo? Due to this, it is not clear how the thresholds of $p_T < 8$ GeV and $p_T < 6$ GeV were selected. Can you please clarify?
- e) The last sentence in the summary is too strong. It also contradicts the previous one, and it should be deleted. A "strong evidence" clearly does not imply a "clear solution to a half-a-century-old puzzle". Moreover, the region where v_2 rises provides 3.1 sigma evidence. It is that region where the parton energy loss is subdominant -- as you also say. I'm not aware that the theory would understand the impact of the parton energy loss on the scaling. To my understanding, the empirical scaling is established mainly in the region of the rise of the $v_2(p_T)$. It is, therefore, appropriate to speak just about the evidence.
- f) Page 1. While ref.[4] is about 20 years old, new calculations using lattice QCD were done. See e.g. <https://arxiv.org/pdf/1708.06667.pdf>, which seems to support a KK molecular configuration contributing to $f_0(980)$ (see section V of that paper). This needs to be mentioned to make the view balanced.

Minor comment:

- o) Page 1. A sentence starting with "It is noteworthy that the collision ..." requires a reference.

Reviewer #3

(Remarks to the Author)

The paper presents convincing results in favor of the $f_0(890)$ interpretation as $q\bar{q}$ state. This conclusion has been obtained based on the study of the elliptic flow constituent quark number scaling. Both, the results itself, as well as the application of the technique for study of the hadronic structure deserve, from my point of view, a publication in Nature Communications.

In general the paper reads very well, though a few points have to be clarified before the publication. Below I have several comments and suggestions, which I hope could be addressed by the authors.

Most importantly, more details have to be provided on the calculations of the second harmonic event plane. At present the paper mentions only that this was done with the help of "HF calorimeter" (without explanation what "HF" means) and that the subevent method was used to calculate the event plane resolution. What is the coverage of "HF calorimeter"? How subevents were defined? What was done to exclude "non-flow" correlations between subevents (e.g. energy leakage)?

As the final conclusion is reached by comparing of the $f_0(890)$ flow to that of other particles, how flow of other particles was calculated? How the flow of other particles calculated using the event plane from HF calorimeter compared with published results obtained with two-particle correlation method?

I would also suggest to consider:

Using v_2 p_T dependence as the default case instead of KE dependence (e.g. Fig.~8 instead of Fig.~3)

In Fig.7 add the corrected results. I understand that it would be a repetition of results shown in Fig. 2, but in this case it would help to see the effect of the correction.

Split the first p_T bin into two. The physics of flow at low transverse momenta and around 2 GeV/c could be very different, and

for the current study it might be useful to have a low p_T cut around 1 GeV.

In view of the above comment, it could be helpful to add to Fig.9 the results corresponding to $p_T > 2$ GeV/c (excluding lowest p_T bin).

Version 1:

Reviewer comments:

Reviewer #1

(Remarks to the Author)

Thank you very much for the updated draft and answers to the comments on the BG v_2 and non-flow related to the multiplicity dependence. I'm satisfied with the replies from the authors and as I have already agreed to go for the publication in the Nature Communication. Thanks again.

Reviewer #2

(Remarks to the Author)

I want to thank the authors for their responses and for updating the paper draft. I'm satisfied with all the answers and updates. In particular, the authors explained the empirical nature of Eq.(3) and modified the text to guide the reader better. The authors explained the origin of data points at high- p_T , which are included in Fig.3. They also explained why the KK channel does not impact the measurement. They softened the statements in the summary and added missing references. I can now recommend the paper draft for publication in Nature Communications.

Reviewer #3

(Remarks to the Author)

I thank the author for considering my comments and providing additional material. I am satisfied with most of the responses, except the two points I reiterate below.

After reading the updated manuscript, I strongly recommend to change the default case for v_2 discussion to $v_2(p_T)$ instead of $v_2(K_E)$. In fact, I find the additional reference to the PHENIX manuscript in support of the K_E scaling as worsening the paper, not improving. First of all the K_E scaling has no theoretical justification, and it is not even clear what the authors mean by "better agreement with the data". Also, what is good at RHIC energies, could be not that good at LHC. Finally, using an "empirically" observed scaling to justify the q - \bar{q} composition of f_0 seems to be a rather weak approach.

In view of the results presented in the "rebuttal" I would recommend to use $p_T > 2$ GeV/c cut for Fig. 9 (excluding lowest p_T bin), and correspondingly update the numbers in the main text. The flow in the Low p_T bin could have very different origin compared to higher p_T bins, and I find no reason to include lowest p_T bin in the fit.

We would like to thank the referees for reviewing this paper and furnishing their reports.

We have carefully considered all comments and applied several changes to the original version of the paper to address the issues raised. Detailed responses to all the comments can be found below.

We are at your disposal for any further clarifications and/or additional information.

REVIEWER COMMENTS

Reviewer #1 (Remarks to the Author):

The authors have measured elliptic event anisotropy of f_0 in pPb collisions at LHC and used the number of constituent quark (NCQ) scaling of v_2 to test the quark content of the f_0 hadron assuming the quark coalescence mechanism for the hadron formation. The method is well established in high-energy heavy-ion physics field, while the NCQ scaling is approximately hold for various mesons and baryons. The presented results on f_0 v_2 measurements are interesting and worth publishing, therefore I would like to suggest this manuscript to be published, after a few more additional information presented in the manuscript or most likely in the method section as listed in the following.

1) Concerning on the invariant mass fitting, while the poorly described region below the mass peak of $\rho(770)$ might not be affecting too much on the final results of f_0 v_2 value, the v_2 values of the residual background as well as f_2 and $\rho(770)$, would need to be treated properly, so I would request authors to provide how other contributions behave especially the residual (and combinatorial) BG v_2 below the f_0 peak. This is especially because the residual BG is fitted by 3rd order polynomial function, I would like to know if this is totally independent polynomial function across the different azimuthal bins or if there is any constraint.

There is no constraint on the polynomial function for each azimuthal bin. We have checked the fits and the backgrounds to ensure all the fits converged properly.

The v_2 of the residual background under the f_0 peak ($0.9 < \text{mass} < 1.05$ GeV) is plotted below together with the f_0 v_2 . We have also plotted the v_2 of all opposite- and same-sign pairs for comparison.

2) The analysis is limited within the multiplicity region in pPb collisions in order to be compared with other similar measurements on other light and strangeness hadrons, it is essential to provide the centrality (or multiplicity) dependence of v_2 measurements especially in the small system, since the non-flow subtraction is quite important in this case, therefore I would request authors to show some multiplicity dependence of this measurements in pPb collisions. It might be better to repeat the same measurements in pp collisions especially for the non-flow subtraction, but the lower multiplicity pPb collisions from the same dataset could be easier and good enough. I would like to see a kind of confirmation of the non-flow subtraction assumption taken from the K0S measurement.

Indeed, we chose to study specific multiplicities to compare the new f_0 data to those for other particles from the same high-multiplicity range. Measurements of f_0 v_2 in other multiplicity bins would not directly address the non-flow-related question raised by the referee, as the low-multiplicity subtraction cannot be performed for f_0 state with the available data. (To employ the low-multiplicity subtraction method, jet-like f_0 -hadron correlations are first required to extract the jet-like near-side correlated yield, a 3-particle correlation when f_0 reconstruction is included. Moreover, such analysis requires subtracting the combinatorial three-hadron contributions, making such a study statistically challenging).

Nevertheless, as was requested, we have extracted the $f_0(980)$ v_2 for the low-multiplicity ($0 < N_{trk}^{offline} < 30$) pPb collisions, as shown in the following plot. For comparison, the K0S v_2 measurements for low-multiplicity events are added to the same figure. The f_0 and K0S v_2 results are comparable, justifying our approach of using K0S measurements to carry out non-flow subtraction for f_0 . In addition, we have done extensive studies of systematic uncertainties related to non-flow subtraction, as described in the manuscript, and remain convinced they adequately cover any possible deficit our chosen approach may incur.

Reviewer #2 (Remarks to the Author):

The authors of the manuscript use a novel method to estimate the quark structure of $f_0(980)$ meson. The method is based on previously published observations showing that elliptic flow (v_2) of mesons and baryons exhibits a number of constituent quarks (n_q) scaling. This empirically established scaling may be understood e.g. within phenomenological models of coalescence as a consequence of hadron formation which proceeds from quarks that are nearby in the phase space. This empirical scaling can then be compared between $f_0(980)$ and other hadrons, allowing to test the hypothesis that $f_0(980)$ contains 2 or 4 quarks ($n_q=2$ or $n_q=4$), which was done in the manuscript. Such a method was applied for the first time, and if established, it may become a valuable tool for understanding the structure of hadronic resonances. The results presented in the manuscript are certainly significant and novel, and they have the potential to significantly advance our understanding of the structure of specific hadron resonances. The topic of the structure of hadron resonances touches on the fundamental open questions connected with strong interaction. The results, therefore, deserve attention from a broader audience.

The experimental methods used in the paper are sound, and statistical analysis is precise. At the same time, I need to raise several important questions that have to be addressed before I can recommend the manuscript for publication in Nature Physics. These are the following:

a) Equation (3). No reference to the theory paper is provided for Eq (3). How is the functional form of Eq (3) connected with the n_q scaling of v_2 discussed earlier in the manuscript (formula $v_n \sim n_q * v_n(p_T/n_q)$)? Reference [41], which is the main theoretical paper used here to advocate the scaling, does not give formula (3). Neither

reference [39], which is the original theoretical proposal for this kind of analysis, does. The presence of the exponential term and "Et" instead of pt should be advocated by theory. Please clarify and provide the reference to relevant papers.

An empirical function provided in Eq. 3 describes the dependence of scaled v_2 on scaled KET well. It satisfies the two limits: at $KET=0$, $v_2=0$ by definition; when KET approaches infinity, v_2 is expected to vanish. This functional form is not related to coalescence per se but rather determined by data. The "quark v_2 " behaves this way, as observed in the data. We have added a clarification "empirical function derived from data" above Eq. 3.

Regarding pT and KET scaling, it was found that KET scaling is better than pT scaling by A. Adare et al. (PHENIX Collaboration), PRL 98, 162301 (2007), so we used KET scaling as our default. We have now noted it in the manuscript, citing the PHENIX paper. We use pT scaling as an estimate of the systematic uncertainty due to the choice of the scaling variable and found the effect was small.

b) Figure 3. Figure quotes "PRL 121 (2018) 082301" for datapoints other than those for $f_0(980)$. Datapoints in the PRL paper, however, do not contain datapoints for K_s meson for $KET/n_q > \sim 2.5$ GeV shown here (compare the upper right panel of Fig.3 in the PRL paper with this Figure). How is that possible, please? Were they published in some other, unquoted paper? Or were they reconstructed within the presented analysis? If so, it is not documented in the manuscript. Can you please explain? Those datapoints are crucial for establishing the 7.7 significance shown here.

Indeed, the PRL figure for NCQ-scaled points only extends to KET/n_q of 3 GeV; however, the full range of measured values was provided in the left panel of the figure, as shown below.

c) Subsection A.2. Analysis works under the assumption that measured tracks are pions. PDG, however, says that the KK channel may constitute something between 20% to 50% of hadronical decay width. The $\pi^+\pi^-$ channel, therefore, does not constitute 100% of the hadronic decay width. How does the result change when taking this into account?

Any KK decay channel f_0 contributions are blended in the combinatorial background of π - π invariant mass distributions. They are, therefore, not included in the f_0 yield we extracted from the π - π decay channel. So, indeed, our measured f_0 yield is only from the π - π decays. However, the absolute yields are not important for v_2 measurements; only the relative yields as functions of the azimuthal angle matter.

d) Page 6, sentence starting "As shown in Fig.3, the NCQ scaling ... ". This sentence and the following sentence speak about p_T/n_q while x-axis is $/n_q$. Is that a typo? Due to this, it is not clear how the thresholds of $p_T < 8$ GeV and $p_T < 6$ GeV were selected. Can you please clarify?

The data are measured in p_T bins. The average p_T of each p_T bin is converted into KET when NCQ scaling is analyzed in KET.

The p_T range thresholds are chosen to be the range in p_T/n_q in which the NCQ scaling of other hadrons holds, so the threshold value depends on the n_q hypothesis for the f_0 .

e) The last sentence in the summary is too strong. It also contradicts the previous one, and it should be deleted. A "strong evidence" clearly does not imply a "clear solution to a half-a-century-old puzzle". Moreover, the region where v_2 rises provides 3.1 sigma evidence. It is that region where the parton energy loss is subdominant -- as you also say. I'm not aware that the theory would understand the impact of the parton energy loss on the scaling. To my understanding, the empirical scaling is established mainly in the region of the rise of the $v_2(p_T)$. It is, therefore, appropriate to speak just about the evidence.

For the tetraquark $n_q=4$ exclusion, the entire $p_T < 10$ GeV/c is relevant because it implies constituent quark $p_T/n_q < 2.5$ GeV/c, within which NCQ scaling holds for other hadrons. So, the tetraquark exclusion significance is indeed 7.7 sigma. For completeness, we provided the tetraquark exclusion significances for the other two p_T ranges so full information is available to the reader.

Nevertheless, we have softened the text from "present a clear solution" to "offer a solution."

f) Page 1. While ref.[4] is about 20 years old, new calculations using lattice QCD were done. See e.g. <https://arxiv.org/pdf/1708.06667.pdf>, which seems to support a KK molecular configuration contributing to $f_0(980)$ (see section V of that paper). This needs to be mentioned to make the view balanced.

We have included this reference.

Minor comment:

o) Page 1. A sentence starting with "It is noteworthy that the collision ... " requires a reference.

We have included the relevant reference: B. Alver et al. Phys. Rev.C 77, 014906 (2008).

Reviewer #3 (Remarks to the Author):

The paper presents convincing results in favor of the $f_0(890)$ interpretation as q - \bar{q} state. This conclusion has been obtained based on the study of the elliptic flow constituent quark number scaling. Both, the results itself, as well as the application of the technique for study of the hadronic structure deserve, from my point of view, a publication in Nature Communications.

In general the paper reads very well, though a few points have to be clarified before the publication. Below I have several comments and suggestions, which I hope could be addressed by the authors.

Most importantly, more details have to be provided on the calculations of the second harmonic event plane. At present the paper mentions only that this was done with the help of "HF calorimeter" (without explanation what "HF" means) and that the subevent method was used to calculate the event plane resolution. What is the coverage of "HF calorimeter"? How subevents were defined? What was done to exclude "non-flow" correlations between subevents (e.g. energy leakage)?

We have spelled out 'HF': "hadron forward (HF) calorimeters" and noted the pseudorapidity coverage of $3 < \eta < 5$ for the one we use.

The EP resolution is calculated with the 3-subevent method,

$$R_n^a\{\Psi_m\} = \sqrt{\frac{\langle \cos [n(\Psi_m^a - \Psi_m^b)] \rangle \langle \cos [n(\Psi_m^a - \Psi_m^c)] \rangle}{\langle \cos [n(\Psi_m^b - \Psi_m^c)] \rangle}}.$$

Here Ψ_a is the EP angle reconstructed from energies deposited in the HF calorimeter at $3 < \eta < 5$ (Pb-going direction), Ψ_b is reconstructed from the HF calorimeter at $-5 < \eta < -3$ (p-going direction), and Ψ_c is reconstructed from particle information measured in the central tracker detector ($|\eta| < 0.75$). The eta gaps between the subevents help to suppress the nonflow effects.

"The resolution is obtained by the subevent method." has been changed to "The resolution is obtained by the three-subevent method." We have also added a sentence: "The three-subevent method uses the two HFs and the central tracker detector, where the eta gaps between the subevents help suppressing the nonflow effects."

As the final conclusion is reached by comparing of the $f_0(890)$ flow to that of other particles, how flow of other particles was calculated? How the flow of other particles calculated using the event plane from HF calorimeter compared with published results obtained with two-particle correlation method?

The flow of other particles was calculated using the two-particle correlation method.

It has been shown that the EP method and the two-particle correlation method are essentially the same. This similarity arises because the EP is reconstructed from all other particles. Therefore, the POI-EP correlation is "equivalent" to the sum of the

correlations of POI to all other particles (POI: particle of interest). For example, the D^0 meson v_2 published by CMS in Phys. Rev. Lett. **120** (2018) 202301 used the same EP method as in this work. The results were compared to those from the two-particle correlation method. The results are consistent with each other, as shown in the plots below.

For the f_0 analysis in this work, we did not use the two-particle correlation method because this is essentially a three-particle correlation (two pions from f_0 to be correlated with a third hadron) and subtraction of the large three-particle combinatorial background.

Figure 63: Comparison of $D^0 v_2$ results from Jing and Jian (main analysis results) for centrality 10-30% and 30-50%

Figure 90: Comparison between D^0 meson v_2 measurement from two-particle correlation and scalar product methods.

I would also suggest to consider:

Using $v_2 p_T$ dependence as the default case instead of KE dependence (e.g. Fig.~8 instead of Fig.~3)

It has been observed that KET scaling is better than p_T scaling, so we prefer to use KET as default. We have added a sentence regarding the rationale to use KET above Eq. 3 with reference to the PHENIX paper: "The KET variable is chosen to describe

the NCQ scaling as it yields better agreement with the data than pT
\cite{Adare:2006ti}".

In Fig.7 add the corrected results. I understand that it would be a repetition of results shown in Fig. 2, but in this case it would help to see the effect of the correction.

We have added the v2sub in Fig.7.

Split the first pT bin into two. The physics of flow at low transverse momenta and around 2 GeV/c could be very different, and for the current study it might be useful to have a low pT cut around 1 GeV.

The highest sensitivity of this measurement is from the high-pT region. The low-pT data points do not provide significant discrimination between the different nq hypotheses. Moreover, the lowest data point does not have great statistical precision, so splitting it into two or more bins would not offer a better insight into the pT dependence. So we prefer to keep the current pT binning.

We have also checked the significance, excluding the first pT bin. The significance values are 7.71, 6.37, and 3.01 (corresponding values, including the first pT bin, are 7.72, 6.33, 3.05.). There is no significant difference.

In view of the above comment, it could be helpful to add to Fig.9 the results corresponding to $p_T > 2$ GeV/c (excluding lowest pT bin).

The plot below shows the Chi-square scan for $2 < p_T < 6$ GeV/c. The curves for $2 < p_T < 8$ GeV/c and $2 < p_T < 10$ GeV/c are practically identical to those for $0 < p_T < 8$ GeV/c and $0 < p_T < 10$ GeV/c, respectively. The chi2 curves are very similar with and without the lowest pT bin because the lowest pT data point falls almost on the fitted NCQ reference curve. We prefer to keep the current version of Fig. 9 with the three original pT ranges.

For reference, we have plotted below the chi2 vs. n_q with/without the first pT bin for different pT ranges. There is no significant difference.

We would like to thank the referees for reviewing this paper and furnishing their reports. We have carefully considered all comments and applied several changes in the revised paper to address the issues raised. Detailed responses to the comments can be found below.

In addition, we have reorganized the text to comply with journal requirements you outlined:

1. Changed the “**The $f_0(980)$ puzzle and relativistic heavy ion collisions**” section name to “**Introduction**”
2. Changed the second to last sentence of the “**Introduction**” section: “*The NCQ scaling of the $f_0(980)$ hadron v_2 coefficient is tested, under the $n_q = 2$ and 4 hypotheses, as well as under the $n_q = 3$ hypothesis, which is expected to be characteristic of a hybrid state.*”

into

“The NCQ scaling of the $f_0(980)$ hadron v_2 coefficient is tested. We demonstrate that the hypothesis of the $f_0(980)$ state being an ordinary $q\bar{q}$ meson is significantly preferred over alternative hypotheses. This novel technique could be used to investigate other exotic hadron candidates.”

3. Moved detector paragraph to Methods section. Added a sentence in the main text to refer to that.
4. Moved trigger paragraph in the original Section 2 to Methods section and merge with the “Trigger” subsection.
5. Figure 6 and 7 merged as a single Figure with two panels (now Figure 6). Caption changed accordingly
6. Figure 10 and 11 merged as a single Figure with three panels (now Figure 9). Caption changed accordingly
7. Methods section moved to the main text, instead of appendix.

We are at your disposal for any further clarifications and/or additional information.

Color code

Black: Referee comment

Blue Author response

Green: Quoted original text

Green: Quoted revised text.

Reviewer #1

Thank you very much for the updated draft and answers to the comments on the BG v2 and non-flow related to the multiplicity dependence. I'm satisfied with the replies from the authors and as I have already agreed to go for the publication in the Nature Communication. Thanks again.

Reviewer #2

I want to thank the authors for their responses and for updating the paper draft. I'm satisfied with all the answers and updates. In particular, the authors explained the empirical nature of Eq.(3) and modified the text to guide the reader better. The authors explained the origin of data points at high-pt, which are included in Fig.3. They also explained why the KK channel does not impact the measurement. They softened the statements in the summary and added missing references. I can now recommend the paper draft for publication in Nature Communications.

Reviewer #3

I thank the author for considering my comments and providing additional material. I am satisfied with most of the responses, except the two points I reiterate below.

After reading the updated manuscript, I strongly recommend to change the default case for v2 discussion to v2(pT) instead of v2(KE). In fact, I find the additional reference to the PHENIX manuscript in support of the KE scaling as worsening the paper, not improving. First of all the KE scaling has no theoretical justification, and it is not even clear what the authors mean by "better agreement with the data". Also, what is good at RHIC energies, could be not that good at LHC. Finally, using an "empirically" observed scaling to justify the q-qbar composition of f0 seems to be a rather weak approach.

On one hand, p_T scaling is more theoretically motivated, such as the coalescence model. On the other hand, the empirical scaling in KE_T is an experimental observation, and thus our results are less theory/model dependent. We pointed out these aspects at the end of the second to last paragraph in Section 1: *"We note that the NCQ scaling may additionally arise from other mechanisms in the same or an extended p_T range. The empirical observations of the NCQ scaling do not depend, however, on a particular underlying physics mechanism."*

Moreover, using p_T scaling does not change our conclusion. We have expanded the last paragraph in Section A.3 (now Section 4.4):

"The NCQ-scaled v_2^{sub}/n_q as a function of p_T/n_q is shown in Fig. 8 for the $f_0(980)$ state together with those of the K_S^0 , Λ , Ξ^- , and Ω hadrons [Phys. Rev. Lett. 121 (2018) 082301]."

into

"The NCQ scaling can also be parameterized in p_T/n_q . The fit quality is worse, with $\chi^2/dof=170/34$. The NCQ-scaled v_2^{sub}/n_q as a function of p_T/n_q is shown in Fig. 7 for the $f_0(980)$ state together with those of the K_S^0 , Λ , Ξ^- , and Ω

hadrons [Phys. Rev. Lett. 121 (2018) 082301]. Using the NCQ scaling in p_T/n_q , the extracted significance of the $n_q = 2$ hypothesis over the $n_q = 4$ hypothesis is 7.8, 6.2, or 2.4σ , in the $p_T < 10, 8, \text{ or } 6 \text{ GeV/c}$ ranges, respectively."

To make the text smoother, we have moved this paragraph and the accompanying Fig. 8 (now Fig. 7) from Section A.3 (now Section 4.4) to Section A.5 (now Section 4.6) as the second paragraph. We added "*(as discussed in the Methods section)*" following the sentence "*The n_q extraction procedure is repeated for variations in the functional form of $f(KE_T/n_q)$, as well as by using p_T instead of KE_T in the NCQ scaling expression given by Eq. (3).*" in the Section 3 (now Section 2.2).

We cited the PHENIX paper to show that it is an experimental observation that the NCQ scaling is better with KE_T than with p_T . This is also true at the LHC as shown in *Phys. Rev. Lett. 121 (2018) 082301*, which we also cited. By the "better agreement", we mean that the χ^2/dof is smaller for the NCQ scaling fit with KE_T ($\chi^2/dof = 80/34$) than that with p_T ($\chi^2/dof = 170/34$). The former χ^2/dof value was already quoted in the paper. We have now included the p_T fit χ^2/dof in the extended paragraph above: "*The fit quality is worse, with $\chi^2/dof = 170/34$.*"

In view of the results presented in the "rebuttal" I would recommend to use $p_T > 2 \text{ GeV/c}$ cut for Fig. 9 (excluding lowest p_T bin), and correspondingly update the numbers in the main text. The flow in the Low p_T bin could have very different origin compared to higher p_T bins, and I find no reason to include lowest p_T bin in the fit.

Since including the data point or excluding it does not really change the result, we prefer to keep it in the fit. To clarify better, we have added a new paragraph before the last paragraph in Section A.5 (now Section 4.6):

"The NCQ scaling may be less valid at low p_T where v_2^{sub} likely reflects hydrodynamic behavior, which is mass dependent. However, excluding the lowest p_T data point has a negligible effect on our results."